# Geographically weighted regression analysis of anemia and its associated factors among reproductive age women in Ethiopia using 2016 demographic and health survey

Daniel Gashaneh Belay[1,2]*, Shumet Mebrat Adane[3], Oshe Lemita Ferede[2], Ayenew Molla Lakew[2]

1 Department of Human Anatomy, College of Medicine and Health Sciences, University of Gondar, Gondar, Ethiopia, 2 Department of Epidemiology and Biostatistics, Institute of Public Health, College of Medicine and Health Sciences, University of Gondar, Gondar, Ethiopia, 3 Department of Epidemiology, College of Medicine and Health Sciences and Comprehensive Specialized Hospital, Hawassa University, Hawassa, Ethiopia

* danielgashaneh28@gmail.com

**Data Availability Statement:** All relevant data are within the paper and its Supporting information files.

## Abstract

### Introduction

Anemia in reproductive age women is defined as the hemoglobin level <11g/dl for lactating or pregnant mothers and hemoglobin level <12 g/dl for none pregnant or non-lactating women. Anemia is a global public health problem affecting both developing and developed countries. Therefore this study aims to determine geographically weighted regression analysis of anemia and its associated factors among reproductive age women in Ethiopia using the 2016 Demographic and Health Survey.

### Method

In this study, a total of 14,570 women of reproductive age were included. Multi-level binary logistic regression models were employed using STATA version 14. Odds ratio with a 95% confidence interval and p-values less than 0.05 was used to identify significant factors. Spatial scan statistics were used to identify the presence of anemia clusters using Kulldorf's SaTScan version 9.6 software. ArcGIS 10.7 software was used to visualize the spatial distribution and geographically weighted regression of anemia among reproductive age women.

### Result

Overall 23.8% of reproductive-age women were anemic. The SaTScan spatial analysis identified the primary clusters' spatial window in Southeastern Oromia and the entire Somali region. The GWR analysis shows that having a formal education, using pills/injectables/ implant decreases the risks of anemia. However, women who have more than one child within five years have an increased risk of anemia in Ethiopia. In addition to these, in multi-level analysis women who were married and women who have >5 family members were more likely to have anemia.

**Funding:** The author(s) received no specific funding for this work.

**Competing interests:** The authors have declared that no competing interests exist.

## Conclusion

In Ethiopia, anemia among reproductive age women was relatively high and had spatial variations across the regions. Policymakers should give attention to mothers who have a low birth interval, married women, and large family size. Women's education and family planning usage especially pills, implants, or injectable should be strengthened.

## Introduction

Anemia in reproductive age women is defined as the hemoglobin level <11 g/dl for lactating or pregnant mothers and hemoglobin level <12 g/dl for none pregnant or non-lactating women or a decline in the concentration of circulating erythrocytes in the blood and a concomitant impairment of oxygen transportation [1]. It affects people at all stages of their lives, but it is more common among young children and pregnant women [1].

Reproductive age is commonly defined among women as ages 15 to 49 years [1]. They are physiologically more prone to anemia as a result of the constant loss of blood during menstruation and the demands of pregnancy and childbearing [2].

Anemia is a worldwide public health problem that affects both developing and developed countries, with serious ramifications for human health, social and economic development [1]. It is an indicator of both poor nutrition and poor health [1]. Anemia is the most frequent and persistent nutritional problem in the world, and one of the primary indirect causes of maternal mortality [2]. It is one of the most serious dangers to children's health and a factor in maternal mortality, because it increases the risk of adverse pregnancy outcomes, child mortality, impaired neurocognitive abilities, and physical development of children, and reduces work capacity despite being straightforward to prevent and treat [1, 3].

Anemia affects 1.62 billion (24.8%) individuals worldwide, with different epidemiology's depending on population age, sex, socio-cultural contexts, and geographical locations [2, 4, 5]. According to the World Health Organization (WHO) report, worldwide approximately 46% of pregnant women and 39% of Women at Reproductive Age (WRA) were affected by anemia [1, 2]. In Africa, based on WHO regional estimates generated for preschool-age children, pregnant and non-pregnant women indicate that the proportion of individuals affected by anemia ranges from 47.5–67.6% which was the highest from other regions of the world [4, 5]. In Ethiopia, the prevalence of anemia in reproductive age women decreased from 27% in 2005 to 17% in 2011 [6]. Moreover, the pooled prevalence of anemia among pregnant women from 2003 to 2016 was 31.66% in Ethiopia [7].

Individual-level factors such as, being pregnant [6], current lactation and those who gave birth in the month of the interview [3], women who gave birth within five years [6], having a large family size [2, 8, 9], and women who live in a household with low wealth index [2, 6] have a positive association with anemia among reproductive age women. Whereas current hormonal contraceptives users such as pills, implants, or injectable [3, 6], prim gravida [7], and women with secondary and above education were [6] had a protective effect on anemia among reproductive age women. Community-level factors such as residence and region have a significant association with anemia among reproductive age women [6, 7].

Recognizing it as a worldwide public health problem, the WHO target is set to reduce anemia in women of reproductive age(WRA) by 50% in 2025 [6]. According to the Ministry of Health of Ethiopia, maternal nutrition is one of the top priorities and the prevalence of anemia in WRA is among the outcome indicators of Health Sector Transformation Plans (HSTP) [6, 10]. The Ethiopian Federal Ministry of Health (FMoH) has been making efforts to prevent

anemia focusing on pregnant women by supplying iron (Fe) and folic acid, proper nutrition, education, deworming, promoting sanitation, and preventing and treating anemia. However, in the last 15 years, the trend of anemia has remained inconsistent [11]. Even though the above intervention has been taken, the prevalence of anemia among reproductive age women in Ethiopia is still high [6, 7].

## Methods

### Study design and setting

The study used population-based cross-sectional survey data from 2016 Demographic Health Surveys conducted in Ethiopia. Ethiopia ($3^0$–$14^0$ N and $33^0$–$48^0$E) is located in the horn of Africa. The country covers 1.1 million Sq. Kilometers, with huge geographic diversity: from 4550m above sea level to 110m below sea level in Afar depression. There are nine regional states(Amhara, Afar, south nation nationality and peoples, Gambela, Benshangul Gumuz, Harari, Oromia, Somalia, and Tigry) and two city administrations (Addis Ababa and Dire Dawa). These areas are divided into 68 zones, 817 districts, and 16,253 kebeles (lowest local administrative units of the country) in the administrative structure of Ethiopia [12].

### Source and study population

The source population was all women aged 15 to 49 within five years before the survey in Ethiopia, while all reproductive-age women in the selected enumeration areas were the study population. EDHS uses a two-stage stratified cluster sampling method, using the 2007 Population and Housing Census as the sampling frame. First, 645 enumeration areas (EA) were chosen with a probability proportionate to their size, and an independent sample was drawn at each sample level. And then 28 households were systematically selected on average. Hemoglobin level was done for 14,489 women and of them, 14,171 women were usually live in the surveyed households (de juries) and included in the study. Therefore, the final analysis in "Fig 1" uses a total weighted sample of 14,570 women. The data collection took place from 18 January 2016 to 27 June 2016.

### Outcome variable

The current study is based on the altitude adjusted hemoglobin levels which were already reported in 2016 EDHS data. Anemia is defined as the hemoglobin level <11 g/dl for lactating or pregnant mothers and hemoglobin level <12 g/dl for none pregnant or non-lactating women [1].

### Independent variables

Individual-level and community-level factors were used. The variables were selected based on the literature review for factors affecting anemia, then sociodemographic, maternal, as well as community-level factors, were identified as important factors for the occurrence of anemia. Individual factors included age, women education, religion, marital status, mass media exposure, alcohol consumption, khat chewing (stimulant plant), current pregnancy, lactating mother, history of abortion, contraceptive method, number of birth in last 5 years, wealth index, family size, cooking fuel, toilet facility, and drinking water source. Community-level factors such as place of residence, region, community poverty, community mass media exposure, and community women education were used. The recoding of community aggregate factors has been taken from national report percentages. For community poverty, according to the world bank (WB), in 2019/2020 report around 24% of the population is under poverty

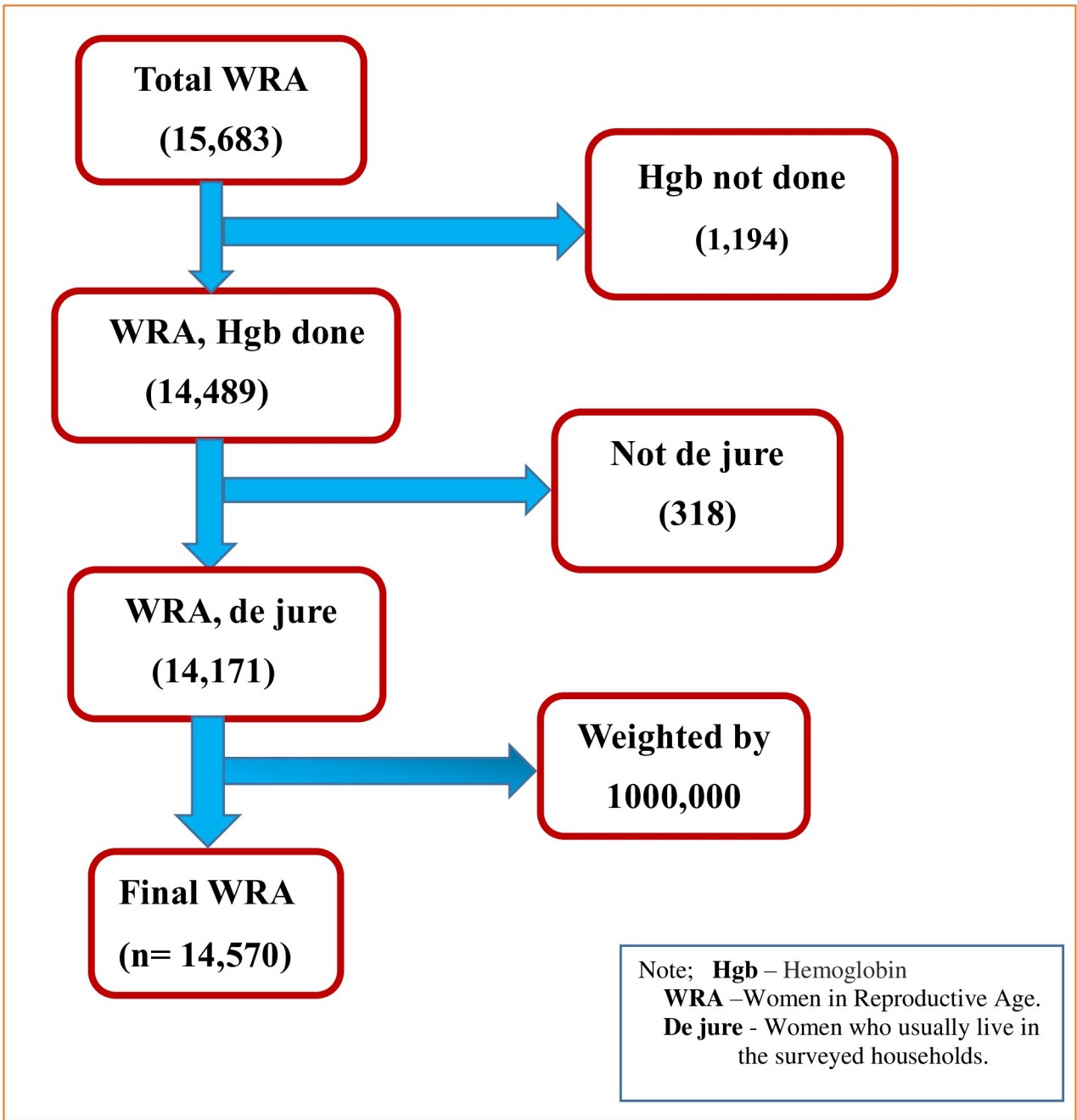

**Fig 1. Sampling procedure and sampling technique geographically weighted regression analysis of anemia and its associated factors among reproductive-age women in Ethiopia using 2016 EDHS.**

[13]. For community mass media exposure we have used 13.8% and also for community women's education level we used 7.7% [6]. The normal distribution of aggregated community factors was assessed by histogram and Shapiro Wilks test but, they didn't fulfill the normality assumption then we recode them based on the median value.

## Data processing and analysis

We accessed the data sets using the website www.measuredhs.com after the rational request of the Demographic and health survey (DHS). The geographic coordinate data (latitude and longitude coordinates) were also taken from selected enumeration areas through the web page of the international DHS program. The required data treatment and cleaning process was made using Stata version 14 statistical software. Descriptive analyses were used to explain the prevalence of anemia among WRA groups. Before performing spatial analysis, the weighted proportion (using sample weight) of anemia among WRA and candidate explanatory variables data were exported to ArcGIS.

## Model building

Due to the hierarchical nature of the 2016 EDHS data, where individuals are nested within the community, the assumptions such as independent of observations and equality of variance have been violated. Therefore multilevel binary logistic regression was fitted for the study of determinants of anemia among reproductive age women. Four models were used in the multilevel analysis. The first model contained only the outcome variable which was used to check the proportion of anemia among WRA variability in the community. The second models contain only individual-level variables and the third model contains only community-level variables, whereas, in the fourth model, both the individual and community-level variables were adjusted simultaneously with the outcome variables. Model comparison was done using the loglikelihood ratio test and the fourth model, which has the highest log-likelihood ratio was selected as the best fit model.

## Parameter estimation method

Both random effect and fixed effect model parameters were included in the model.

Random-effects estimates the variation of prevalence of anemia among reproductive age women between clusters. We used the cluster number variable (v001) for random effect estimates. We estimated the intraclass correlation coefficient (ICC), the median odds ratio (MOR), and Proportional Change in Variance (PCV). The intraclass correlation coefficient (ICC) reveals that, the variation of anemia among reproductive age women due to the cluster difference. $ICC = \frac{VA}{VA + 3.29} * 100\%$, where;

VA = area/cluster level variance [14–16].

The MOR can be understood as the increased risk (in median) that would have if moving to another area with a higher risk [16].

MOR = exp.$[\sqrt{(2 \times VA)} \times 0.6745]$, or MOR = $e^{0.95\sqrt{VA}}$ where; VA is the area level variance [14, 16].

The PCV reveals the variation in anemia among reproductive age women which is explained by all factors. The PCV is calculated as; $PCV = \frac{Vnull - VA}{V\ null} * 100\%$ where; Vnull = variance of the first model, and VA = variance of the model with more terms [14, 16].

The fixed effect assesses the relationship between the possibilities of anemia among women of reproductive age and predictors. For the final model, factors with a p-value ≤ of 0.2 in crude odds ratio (COR) were selected. Associations between outcome and explanatory variables were assessed and its strength was presented using adjusted odds ratios with 95% confidence intervals with a P-value of <0.05 cut point.

## Spatial analysis

For spatial analysis, Arc GIS 10.7 and SaTScan version 9.6 software were used. A statistical measurement of spatial autocorrelation (Global Moran's I) is used for the assessment of the spatial distribution of anemia among WRA in Ethiopia [17]. Hot Spot Analysis (Getis- Ord Gi* statistic) represents the cluster characteristics with hot or cold spot values spatially. Whereas the ordinary Kriging spatial interpolation technique is used to predict the proportion of anemia among WRA for unsampled areas in the country based on sampled EAs. Bernoulli-based model spatial scan statistics were employed to determine the geographical locations of statistically significant clusters for the prevalence of anemia among WRA. To fit the Bernoulli model, cases were taken from the scanning window that moves across the study area in which women had anemia, and controls were taken from those women who had no anemia. The default maximum spatial cluster size of < 50% of the population was used as an upper limit, allowing both small and large clusters to be detected. The primary, secondary, and other significant clusters were identified and ranked based on the likelihood ratio test (LLR) test using 999 replications of Monte Carlo. The circle with the highest statistic in the LLR test is defined as the most likely (primary) clusters, that is, the group with the least random occurrence.

## Ordinary least square analysis

The ordinary least square analysis was done using variables that were found to be significant at the final multilevel model. The Ordinary Least Square regression (OLS) model is a global model that predicts only one coefficient per independent variable over the entire research area. Then, the model performance, as well as the model significance such as VIF, R-square, Koenker, and Jarque-Bera statistics, expected sign for coefficients, and spatial autocorrelation of residuals were checked.

The model structure of ordinary least square analysis equation [18] is written as,

$$Yi = \beta o + \sum_{k=1}^{p} \beta k \, Xik + \varepsilon i$$

where $i = 1, 2, \ldots n$; $\beta_0, \beta_1, \beta_2, \ldots \beta_p$ are the model parameters, $y_i$ is the outcome variable for observation $i$, $x_{ik}$ are explanatory variables and $\varepsilon_1, \varepsilon_2, \ldots \varepsilon_n$ are the error term/residuals with zero mean and homogenous variance $\sigma^2$

## Geographically weighted regression analysis

Unlike OLS that fits a single linear regression equation to all of the data in the study area, GWR creates an equation for each coefficient.

The model structure of geographically weighted regression equation [19] is written as,

$$Yi = \beta o(ui, vi) + \sum_{k=1}^{p} \beta k(ui, vi) \, Xik + \varepsilon i$$

where $y_i$ is observations of response y, $(u_i v_i)$ are geographical points (longitude, latitude), $\beta_k(u_i, v_i)$ $(k = 0, 1, \ldots p)$ are $p$ unknown functions of geographic locations $(u_i v_i)$, $x_{ik}$ are explanatory variables at the location $(u_i, v_i)$, $i = 1, 2, \ldots n$ and $\varepsilon_i$ are error terms/residuals with zero mean and homogenous variance $\sigma^2$. The OLS and GWR models were compared using different parameters. Finally, the coefficients which were created using GWR were mapped.

## Ethical considerations

The permission for access to the data was obtained from ICF International by registering and stating the purposes of the study. The data set has no household addresses or individual

**Table 1. Prevalence of anemia among WRA with sociodemographic characteristics, 2016 EDHS.**

| Explanatory variable | | Anemia status | | Total |
|---|---|---|---|---|
| | | Yes (%) | No (%) | |
| Age category | 15–19 | 608(19.9) | 2,448 (80.1) | 3,056 |
| | 20–24 | 615(24.3) | 1,919 (75.7) | 2,534 |
| | 25–29 | 681(24.5) | 2,097 (75.5) | 2,778 |
| | 30–34 | 589(26.8) | 1,612(73.2) | 2,201 |
| | 35–39 | 439(24.2) | 1,373(75.7) | 1,812 |
| | 40–44 | 310(25.4) | 911(74.6) | 1,221 |
| | 45–49 | 223(23.0) | 745(77.0) | 968 |
| Religion | Protestant | 844 (24.4) | 2,623(75.6) | 3,467 |
| | Orthodox | 1136(18.2) | 5,094 (81.8) | 6,230 |
| | Muslim | 1366(30.0) | 3,189(70.0) | 4,455 |
| | Others | 118(37.4) | 198(62.6) | 316 |
| Marital status | Not married | 978(18.9) | 4,182(81.1) | 5,160 |
| | Married | 2486(26.4) | 6,923(73.6) | 9,409 |
| Media exposure | No | 2196 (26.3) | 6,139(73.7) | 8,335 |
| | Yes | 1269(20.4) | 4,966(79.6) | 6,235 |
| Alcohol drink | No | 2509(26.5) | 6,949(73.5) | 9,458 |
| | Yes | 956(18.7) | 4,156(81.3) | 5,112 |
| Khat chewing | No | 3029(23) | 9773(76) | 12802 |
| | Yes | 435(25) | 1332(75) | 1767 |
| Breastfeed | No | 2162(21.6) | 7,836(78.4) | 9,998 |
| | Yes | 1302(28.5) | 3, 270(71.5) | 4,572 |
| Contraceptive use | Not using | 2725(25.1) | 8,114 (74.9) | 10,840 |
| | Pills/injectable/implant | 635(18.8) | 2,738 (81.2) | 3,373 |
| | IUCD | 63(30) | 148(70) | 212 |
| | Non-hormonal | 40(27.9) | 104(72.1) | 144 |
| pregnancy | No | 3158(23.4) | 10,343 (76.6) | 13,501 |
| | Yes | 307(28.7) | 762 (71.3) | 1,069 |
| No.birth<5 years | No birth | 1448(19.6) | 5,922(80.4) | 7,369 |
| | One child | 1047(24.1) | 3,319 (76.0) | 4,366 |
| | More than one child | 969(34.1) | 1,865(65.9) | 2,834 |
| Wealth index | Poor | 1523(29.6) | 3,612(70.4) | 5,135 |
| | Middle | 676(23.9) | 2,154(76.1) | 2,829 |
| | Rich | 1266(19.0) | 5,339(81.0) | 6,605 |
| Family size | < = 2 | 194(15,7) | 1,040(84.3) | 1,234 |
| | 3&4 | 870(21.6) | 3,158(78.4) | 4,028 |
| | > = 5 | 2400(25.8) | 6,907(74.2) | 9,307 |
| Cook fuel | Clean | 147(15.6) | 795(84.4) | 942 |
| | Solid | 3317(24.3) | 10,31 1(75.6) | 13,628 |
| Toilet facility | Unimproved | 3040(24.3) | 9,444(75.7) | 12,485 |
| | Improved | 424(20.3) | 1,661(79.7) | 2,085 |
| Drinking water | Unimproved | 1406(28.0) | 3,593(72.0) | 4,998 |
| | Improved | 2059(21.5) | 7,5713(78.5) | 9,572 |
| Hx abortion | No | 3192(24) | 10,204(76) | 13396 |
| | Yes | 273(23) | 902(77) | 1175 |

(*Continued*)

**Table 1.** (Continued)

| Explanatory variable | | Anemia status | | Total |
|---|---|---|---|---|
| | | Yes (%) | No (%) | |
| Women education | No education | 1986(28) | 5116(72) | 7102 |
| | Primary | 1101(21) | 4029(78) | 5130 |
| | Sec & above | 377(16) | 1961(84) | 2338 |
| **Community-level factors** | | | | |
| Residence | Urban | 906(20.33) | 3,549(79.67) | 4,455 |
| | Rural | 2501(25.33) | 7,374(74.67) | 9,875 |
| Region | Tigray | 206(20.01) | 826(79.99) | 1,032 |
| | Afar | 52(44.33) | 65(55.67) | 117 |
| | Amhara | 597(17.55) | 2,807(82.45) | 3,404 |
| | Oromia | 1413(27.13) | 3,797(72.87) | 5,210 |
| | Somalia | 247(59.67) | 167 (40.33) | 414 |
| | B/gumiz | 27(19.15) | 116 (80.85) | 143 |
| | SNNPR | 696(22.67) | 2,375(77.33) | 3,071 |
| | Gambella | 11(27.79) | 30 (73.67) | 41 |
| | Harare | 9(27.79) | 23(72.21) | 32 |
| | Addis Ababa | 125(15.81) | 667 (84.19) | 792 |
| | Derie Dewa | 22(30.13) | 52 (69.87) | 74 |
| Community poverty | Low | 519(18) | 2324 (82) | 2,847 |
| | High | 2946(25) | 8782(75) | 11,728 |
| Community media usage | Low | 431(31) | 964(69) | 1,395 |
| | High | 3033(23) | 10141(77) | 13,174 |
| Community education | Low | 292(42) | 402 (58) | 694 |
| | High | 172(23) | 10704(77) | 13,876 |

names. The data were used for the registered research topic only and were not shared with other subjects. All the data were fully anonymized before we accessed them and/or the ICF International waived the requirement for informed consent. There were no medical records used in the research since it was a demographic and health survey.

## Results

### Sociodemographic characteristics

From the total weighted 14570 reproductive-age women, the mean ± standard deviation (SD) of the respondents' age was 28 ± 9 years. 4,572 (31.38%) women were lactating and 1,069 (7.34%) were pregnant. More than a quarter 25.6% of the respondents were current contraceptive users "Table 1".

### The trend of anemia among reproductive age women in Ethiopia

The prevalence of anemia among reproductive-age women decreased from 27% in 2005 to 17% in 2011, but it increased to 23.78% in 2016 "Fig 2". According to 2016 EDHS, 23.8% [95% CI: 22.7%, 24.8%] of reproductive age women were anemic "Fig 3".

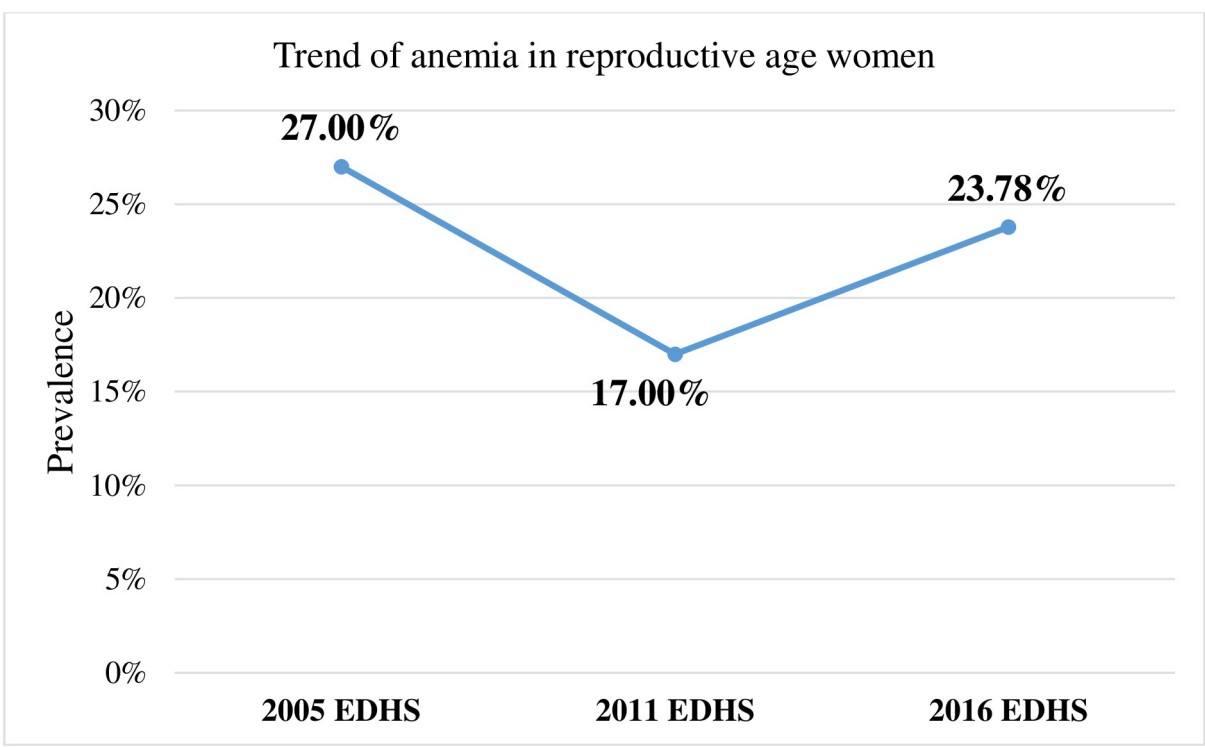

**Fig 2. The line graph on the trend of anemia in reproductive age women from 2005–2016.**

## Multi-level analysis of factors associated with anemia among reproductive age women

**Random effect and model comparison.** The ICC value in the null model of Table 2 showed that 18% of variations of anemia among reproductive-age women were expressed by cluster level factors. The MOR value in the null model also showed that anemic among reproductive-age women were different by 2.28 times between higher and lower prevalence clusters. Moreover, the final model PCV value showed that both the community and individual level factors explained about 40.2% of the variation of anemia among WRA. The deviance and likelihood ratio tests were used to compare and fit the models, and the model with the lowest deviance value and the highest likelihood ratio value which mean Model 4 was the better-fitted model "Table 2".

**Fixed effect outputs.** In multi-level analysis outputs of the final model (model 4), variables such as marital status of women, education status of women, wealth index of the household, family size, hormonal contraceptive usage, and region they live had a significant association with anemia among reproductive age women.

The odds of developing anemia among reproductive age women who were married are 1.2 times that of unmarried women [AOR = 1.23; 95%CI; 1.0, 1.40]. The odds of developing anemia among reproductive age women who were living in rich households are decreased by 16% when compared to poor households [AOR = 0.84; 95%CI: 0.73, 0.96]. Reproductive age women who attended primary and more than primary education were 15% and 24% less likely to have anemia when compared to uneducated one [AOR = 0.85; 95%CI: 0.76, 0.95] & [AOR = 0.76;95%CI: 0.66, 0.94] respectively. The odds of developing anemia among mothers who were using hormonal contraceptives such as pills, injectable, and implants are 26% lower than those not using contraceptives [AOR = 0.74;95%CI: 0.66, 0.83] "Table 3".

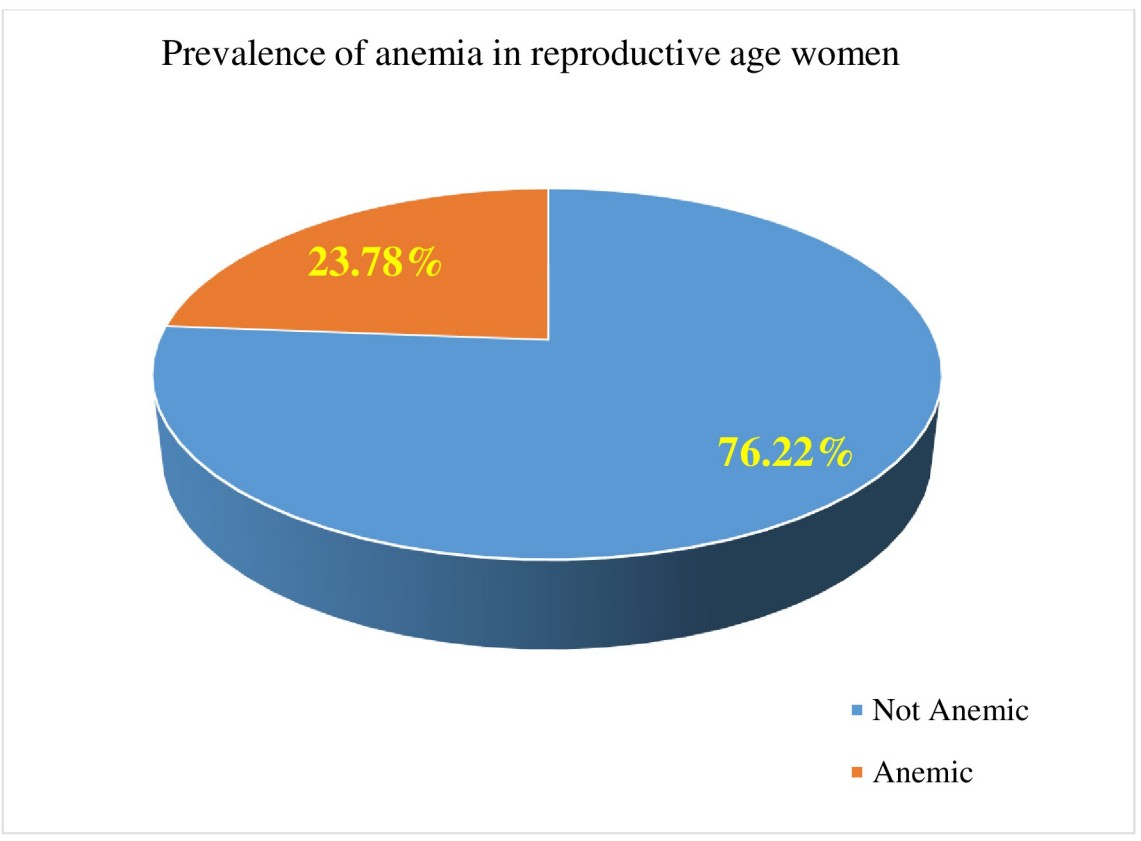

**Fig 3. Pie chart which shows the prevalence of anemia in reproductive-age women, EDHS 2016.**

## Spatial analysis results of anemia among reproductive-age women in Ethiopia (EDHS 2016)

**Spatial distribution, incremental and spatial autocorrelation analysis.** The spatial distribution of anemia among reproductive-age women in Ethiopia shows significant spatial variation across the country. In Afar, Somali, and Dere Dewa regions have a high prevalence of anemia among WRA whereas B/gumiz, Amhara and SNNPR region had low prevalence "Fig 4". Anemia among reproductive-age women was shown to be spatially clustered in Ethiopia,

**Table 2. Parameters and model fit statistics for multi-level models.**

| Parameters | Model 1 (Null) | Model 2 | Model 3 | Model 4 |
|---|---|---|---|---|
| Coefficient variance | 0.72 | 0.54 | 0.44 | 0.43 |
| ICC | 18% | 14.1% | 11.8% | 11.7% |
| MOR | 2.28 | 2.01 | 1.87 | 1.86 |
| PCV | Reff | 25% | 38.9% | 40.2% |
| **Model fitness** | | | | |
| Deviance | 15026 | 14690 | 14806 | 14594 |
| likelihood ratio | | M1&M2 = -112.06 | M2&M3 = 20.94 | M3&M4 = 198.33 |

ICC = Inter cluster corrolation cofficent, MOR = Median odds ratio, PCV = proportional change in varianc

**Table 3. Multi-level analysis factors associated with anemia among reproductive-age women in Ethiopia, from 2016 EDHS.**

| Explanatory variable | | Model–2 | Model–3 | Model–4 |
|---|---|---|---|---|
| | | AOR, [95% CI] | AOR, [95%CI] | AOR, [95%CI] |
| Age category | 15–19 | 1.00 | - | 1.00 |
| | 20–24 | 1.09 [0.94, 1.28] | - | 1.10 [0.94,1.29] |
| | 25–29 | 0.96 [0.83, 1.16] | - | 0,99 [0.84, 1.17] |
| | 30–34 | 1.00 [0.84,1.19] | - | 1.02 [0.86, 1.22] |
| | 35–39 | 0.94 [0.79, 1.13] | - | 0.97 [0.81, 1.16] |
| | 40–44 | 1.05 [0.87, 1.28] | - | 1.07 [0.88, 1.31] |
| | 45–49 | 0.95 [0.76, 1.17] | - | 0.98 [0.78, 1.21] |
| Religion | Protestant | 1.00 | - | 1.00 |
| | Orthdox | **0.82 [0.68, 0.98]**\* | - | **0.80 [0.66, 0.97]**\* |
| | Muslim | **1.40 [1.17, 1.67]**\*\*\* | - | 1.06 [0.87, 1.28] |
| | Others | 0.84 [0.59, 1.22] | - | 0.84 [0.59, 1.21] |
| Marital status | Not married | 1.00 | - | 1.00 |
| | Merried | **1.25 [1.10, 1.44]**\*\* | - | **1.23 [1.07, 1.40]**\*\* |
| Media exposure | No | 1.00 | - | 1.00 |
| | Yes | 0.91 [0.82, 1.02] | - | 0.91 [0.82, 1.01] |
| Alcohol drink | No | 1.00 | - | 1.00 |
| | Yes | 0.92 [0.79, 1.07] | - | 0.97 [0.83, 1.13] |
| Breastfeed | No | 1.00 | - | 1.00 |
| | Yes | 1.00 [0.88, 1.14] | - | 1.03 [0.90, 1.17] |
| Contraceptive use | Not using | 1.00 | - | 1.00 |
| | Pills/injectable/ implant | **0.71 [0.63, 0.80]**\*\*\* | - | **0.74 [0.66, 0.83]**\*\*\* |
| | IUCD | 1.25 [0.89, 1.74] | - | 1.29 [0.92, 1.80] |
| | Non-hormonal | **1.50 [1.00, 2.24]**\* | - | 1.1 [0.79, 1.73] |
| pregnancy | No | 1.00 | - | 1.00 |
| | Yes | 1.04 [0.87, 1.23] | - | 1.06 [0.89, 1.26] |
| No.birth<5 years | No birth | 1.00 | - | 1.00 |
| | One child | 1.08 [0.94, 1.24] | - | 1.06 [0.92, 1.22] |
| | More than one child | **1.43 [1.20,1.69]**\*\*\* | - | **1.36 [1.15, 1.62]**\*\* |
| Wealth index | Poor | 1.00 | - | 1.00 |
| | Middle | 0.97 [0.85, 1.09] | - | 1.01 [0.83, 1.14] |
| | Rich | **0.81 [0.71, 0.92]**\*\* | - | **0.84 [0.73, 0.96]**\* |
| Family size | < = 2 | 1.00 | - | 1.00 |
| | 3&4 | 1.18 [0.97, 1.42] | - | 1.21 [0.94, 1.41] |
| | > = 5 | **1.38 [1.14, 1.65]**\*\* | - | **1.40 [1.17, 1.69]**\*\*\* |
| Cook fuel | clean | 1.00 | - | 1.00 |
| | Solid | 1.20 [0.95, 1.52] | - | 1.23 [0.96, 1.58] |
| Toilet facility | unimproved | 1.00 | - | 1.00 |
| | Improved | 1.09 [0.92, 1.27] | - | 0.96 [0.81, 1.15] |
| Drinking water | Unimproved | 1.00 | - | 1.00 |
| | Improved | 0.93 [0.83, 1.04] | - | 0.95 [0.85, 1.07] |
| Women education | No education | 1.00 | | 1.00 |
| | Primary | 0.83[0.74,0.93] | | **0.85[0.76, 0.95]**\* |
| | secondary&above | **0.78[0.66,0.94]**\*\* | | **0.76[0.66, 0.94]**\* |
| Community-level factors | | | | |
| Residence | Urban | - | 1.00 | 1.00 |
| | Rural | - | **1.69 [1.19, 2.39]**\*\* | 1.32 [0.93, 1.88] |

(*Continued*)

**Table 3.** (Continued)

| Explanatory variable | | Model–2 | Model–3 | Model–4 |
|---|---|---|---|---|
| | | AOR, [95% CI] | AOR, [95%CI] | AOR, [95%CI] |
| Region | Tigray | - | 1.00 | 1.00 |
| | Afar | - | **3.2 [1.98, 5.4]***** | **2.16 [1.27, 3.66]***** |
| | Amhara | - | 0.80 [0.55,1.01] | 0.74 [0.54, 1.00] |
| | Oromia | - | 1.37 [1.03, 1.82]* | 1.03 [0.76, 1.48] |
| | Somalia | - | **5.91 [3.99, 8.76]***** | **3.67 [2.38, 5.65]***** |
| | B/gumiz | - | 0.88 [0.52, 1.49] | 0.69 [0.40, 1.19] |
| | SNNPR | - | 0.97 [0.71, 1.31] | 0.79 [0.56, 1.15] |
| | Gambella | - | 1.55 [0.69, 3.44] | 1.23 [0.85, 2.76] |
| | Harare | - | 1.76 [0.73, 4.22] | 1.29 [0.52, 3.19] |
| | Addis Ababa | - | 1.07 [0.71,1.62] | 1.03 [0.67, 1.58] |
| | Dire Dewa | - | **2.15[1.16, 3.99]*** | 1.56 [0.82, 2.97] |
| Com. education | Low | | 1.00 | |
| | High | | 0.82[0.57,1.17] | 0.93[0.64,1.35] |
| Com.poverty | Low | - | 1.00 | 1.00 |
| | High | - | 0.98 [0.68, 1.42] | 0.71[0.49,1.04] |
| Com. Media | Low | - | 1.00 | 1.00 |
| | High | - | 0.94 [0.71, 1.24] | 0.99 [0.75, 1.31] |

AOR = adjusted odds ratio, CI = confidence interval, IUCD = intrauterine contraceptive device, Com. Media = community media usage, Com. Poverty = community poverty status; Com.education = community education status.

* = P-value < 0.05,

** = Pvalue < 0.01,

*** = Pvalue < 0.001

with a Global Moran's I value of 0.38 (p 0.001). The Z-score of 23.35 indicated that there is less than 1% likelihood that this clustered pattern could result from random chance "Fig 5". The peak distance with statistically significant z-scores on which spatial processes promoting clustering are most pronounced indicated at 151.4 Km; 20.82(distances; Z-score) and 195.8Km; 21.25 (distances; Z-score) "Fig 6".

**Hot and cold spot analysis.** The figure below showed that, the more intense clustering of high (hot spot) proportion anemia among reproductive age women which represent by the red dots. It was clustered at the Somali, Dire Dewa, and Afar regions of Ethiopia. Whereas, Amhara, SNNPR, and Tigray regions of Ethiopia were fewer risk areas which represents by blue dots "Fig 7".

**Spatial sat scan analysis.** There were most likely primary and secondary significant clusters of anemia among WRA. There were a total of 198 significant clusters found, 50 of these were the most probable primary clusters, whereas the remaining 43 were secondary clusters. The primary clusters' spatial window was located in the Somali, Southeastern Oromia region which was centered at 6.023458 N, 44.807507 E with 462.80 km radius, and Log-Likelihood ratio (LLR) of 206.7, at p < 0.001. It showed that in the primary clusters women within the spatial window had 2.33 times higher risk of anemia than women outside the window whereas in the secondary cluster it was 2.37 times higher risk "Table 4" and "Fig 8".

**Kriging interpolation.** Using ordinary kriging interpolation of anemia among reproductive-age women, continuous images have been produced. The predicted anemia among reproductive-age women over the area increases from green to red-colored, which means the red

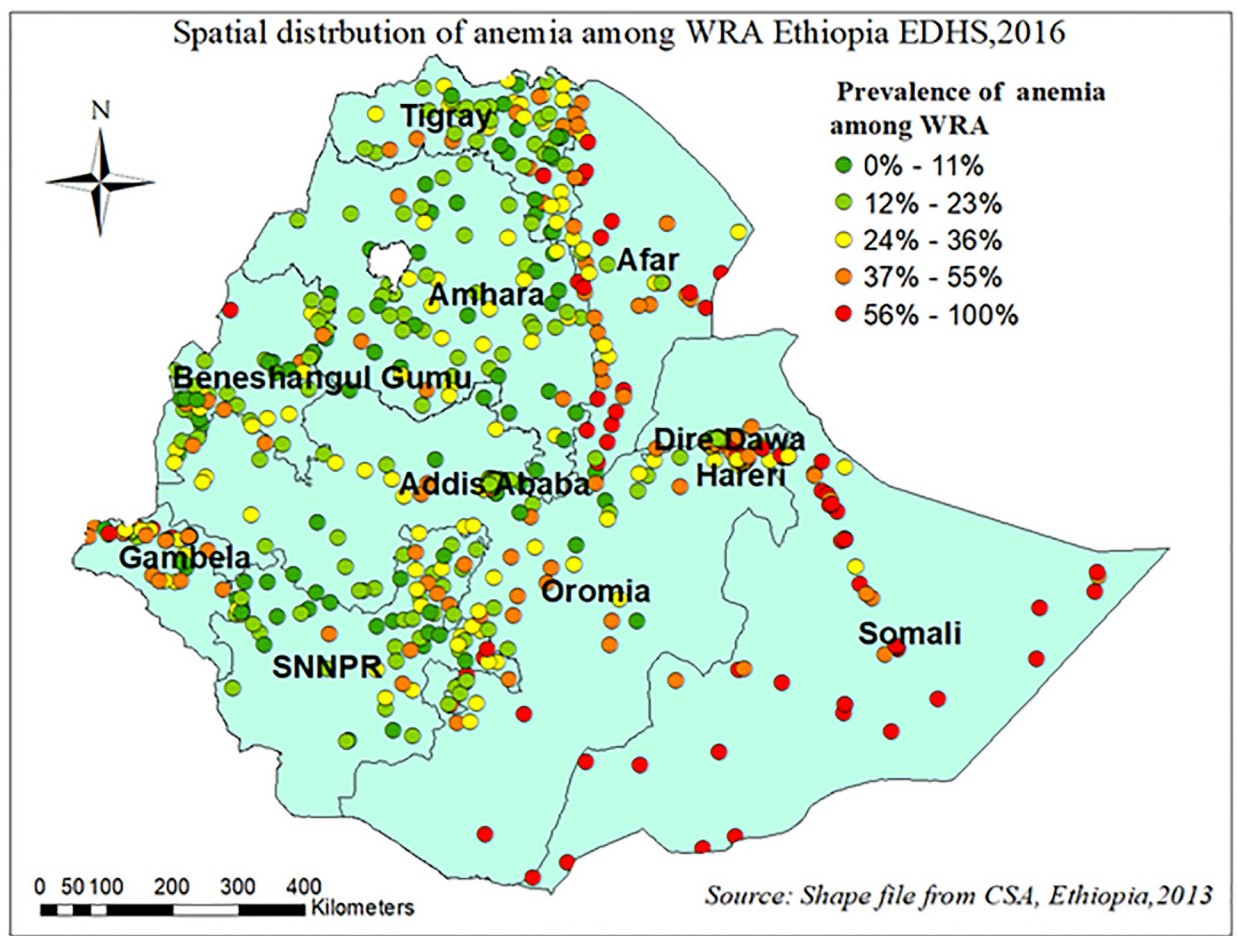

**Fig 4. Spatial distribution of anemia among reproductive age women in Ethiopia, EDHS 2016.**

color indicates high-risk areas of predicted, and the green color indicates the predicted low-risk area of anemia among reproductive-age women. Based on this Somali, Afar and southern parts of the Oromia regions were predicted as riskier than other regions "Fig 9".

## Factors affecting the spatial variation of anemia among reproductive-age women (modeling spatial relationships)

**Ordinary least square regression (OLS).** As shown in Table 5 the OLS model explained about 35.5% (Adjusted R square = 0.355) of the spatial variation in anemia among reproductive-age women. The coefficients represent the strength and the type of each explanatory variable and the anemia WRA Since the Koenker (BP) statistic was significant, we used the robust probability to determine the statistical significance of the coefficients and the coefficients of women who have formal education, women who use Pills/injectable/implant and women have more than one child within five years were statistically significant ($p < 0.01$). The Joint Wald statistic was statistically significant ($p < 0.01$) and this shows that the overall model was significant and also there is no multicollinearity between explanatory variables (Variance inflation factor (VIF) $< 7.5$). In addition, the Spatial Autocorrelation test (Moran's I = 0.21, $P < 0.01$) revealed that residuals were spatially autocorrelated "Table 5".

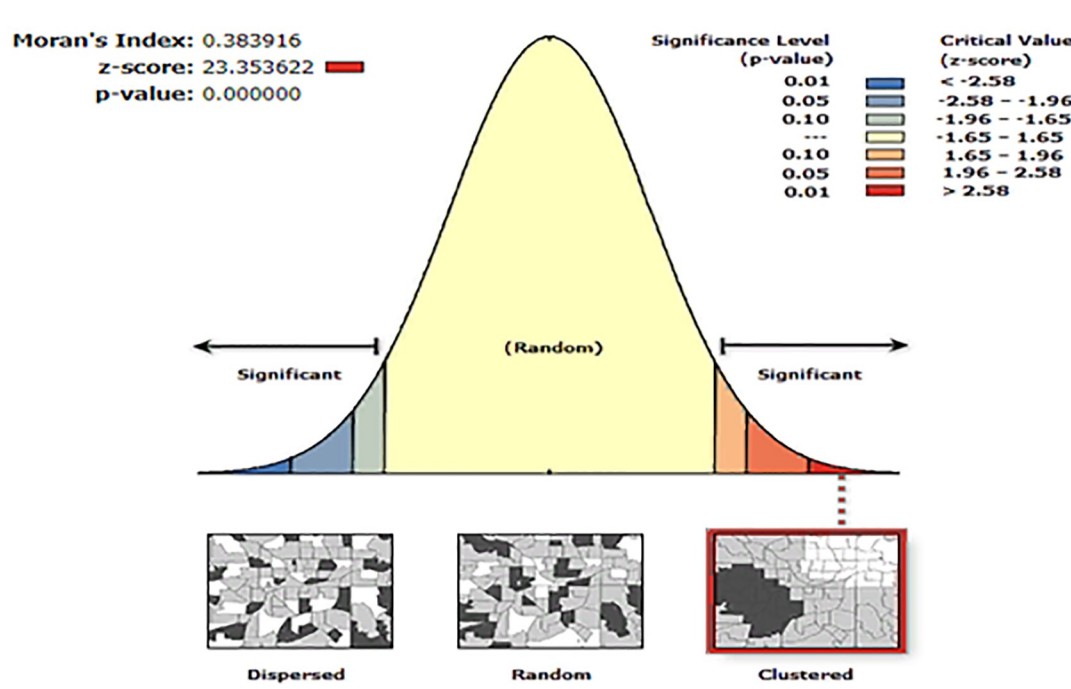

**Fig 5. Spatial autocorrelation of anemia among reproductive age women in Ethiopia, EDHS 2016.**

**Geographically weighted regression.** GWR improves the OLS global model in the case of nonstationarity between predictors and anemia among WRA.

As shown in Tables 5 and 6, the higher the adjusted R square, the lower Akaike's Information Criterion (AICc) value obtained from the GWR model (as compared to the OLS model) helps us to move from a global model (OLS) to a local regression model (GWR). That is conducting the GWR improves the model "Tables 5 and 6".

Fig 10 revealed the model performance (local R square) in which, it was well explained in southern and western parts of Afar, western parts of the Amhara, Dire Dewa, and eastern parts of Somalia regions "Fig 10".

Figs 11–13 demonstrate the geographical areas where the explanatory variables (Attending formal education, using pills/injectable/implant contraceptives, and having more than 1 child within five years) were strong and weak predictors of anemia among WRA in Ethiopia.

Being mothers with formal education had a negative relationship with anemia among WRA. The red-colored clustered points (found in western parts of Amhara, SNNPR, and entire Gambela) indicate areas where the coefficients were largest, which in turn indicates the strong negative relationship between attending formal education and anemia among WRA "Fig 11".

As shown in Fig 12 mothers who use pills, injectable contraceptives and implants showed a strong negative relationship with anemia among WRA in northern Tigray, northern and eastern SNNPR region, and Addis Abeba "Fig 12".

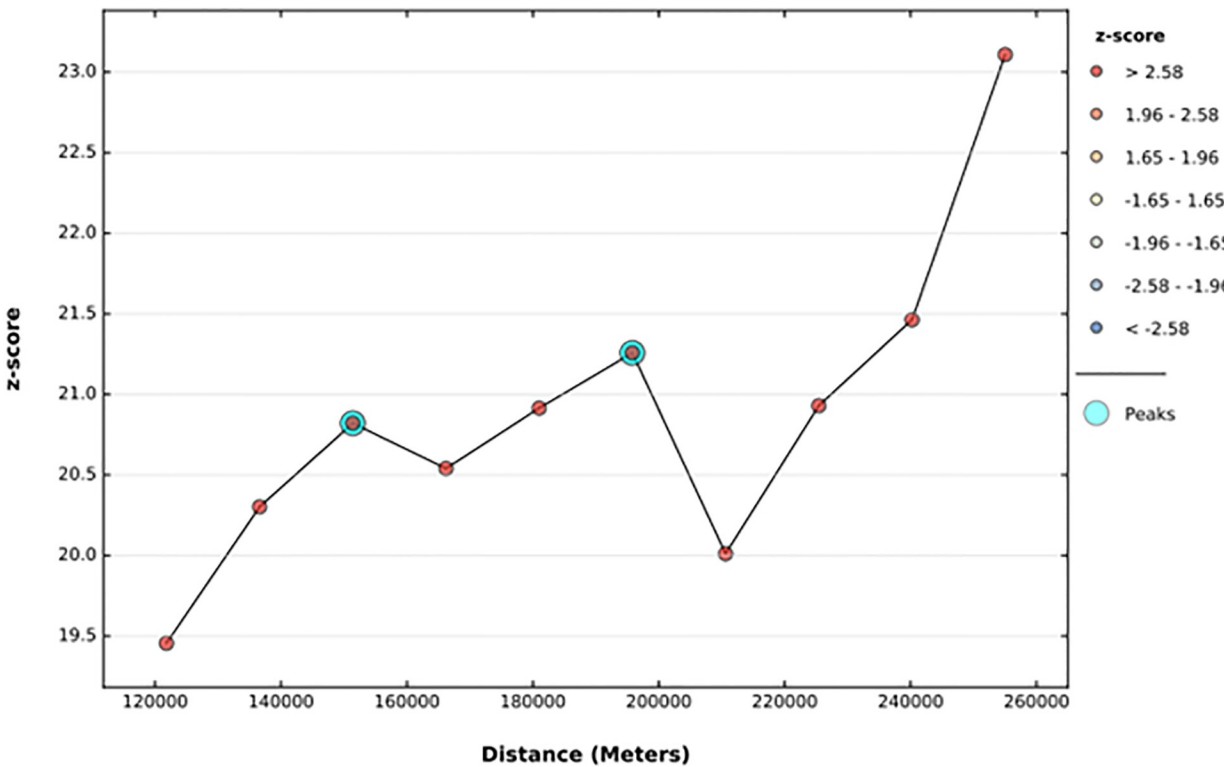

**Fig 6. The incremental autocorrelation of anemia among reproductive age women in Ethiopia by a function of distance using Ethiopian demographic and health surveys 2016.**

Women who have more than one child within five years have a positive relationship with anemia among WRA in eastern Amhara, western Afar, and Somalia region "Fig 13".

## Discussion

Because of their high demand for iron during pregnancy, lactation, monthly bleeding, and nutritional deficiencies, anemia is a serious public health problem among reproductive-age women [2, 10]. This study investigated the prevalence and related factors of anemia in women of reproductive age in Ethiopia evidence on EDHS 2016 using geographically weighted regression analysis.

According to this data, the prevalence of anemia among reproductive-age women was 23.8% [95%CI: 22.7%, 24.8%]. This is in line with a study conducted at Saint Adjibar Ethiopia [20], but lower than a study conducted in developing countries (46.8%) [21], East Africa (34.85%) [22], Uganda (32%) [23], Tanzania (37.6%) [24], seven South and Southeast Asia countries (52.5%) [2], and Nepal (41%) [25]. On the other hand, this study is higher than a study conducted in Rwanda (19.2%) [26]. This difference may be due to geographical disparities, dietary-related factors, socioeconomic level, access to health care, and utilization differences between countries.

When this result is compared with the previous EDHS report, it is lower than 2005 EDHS (27%) but higher than 2011 EDHS(17%) reports [27]. This might be due to the difference in intervention approaches and performance taken by the Ethiopian government. Moreover, the number of reproductive age women included in each EDHS might have its effect [10, 27].

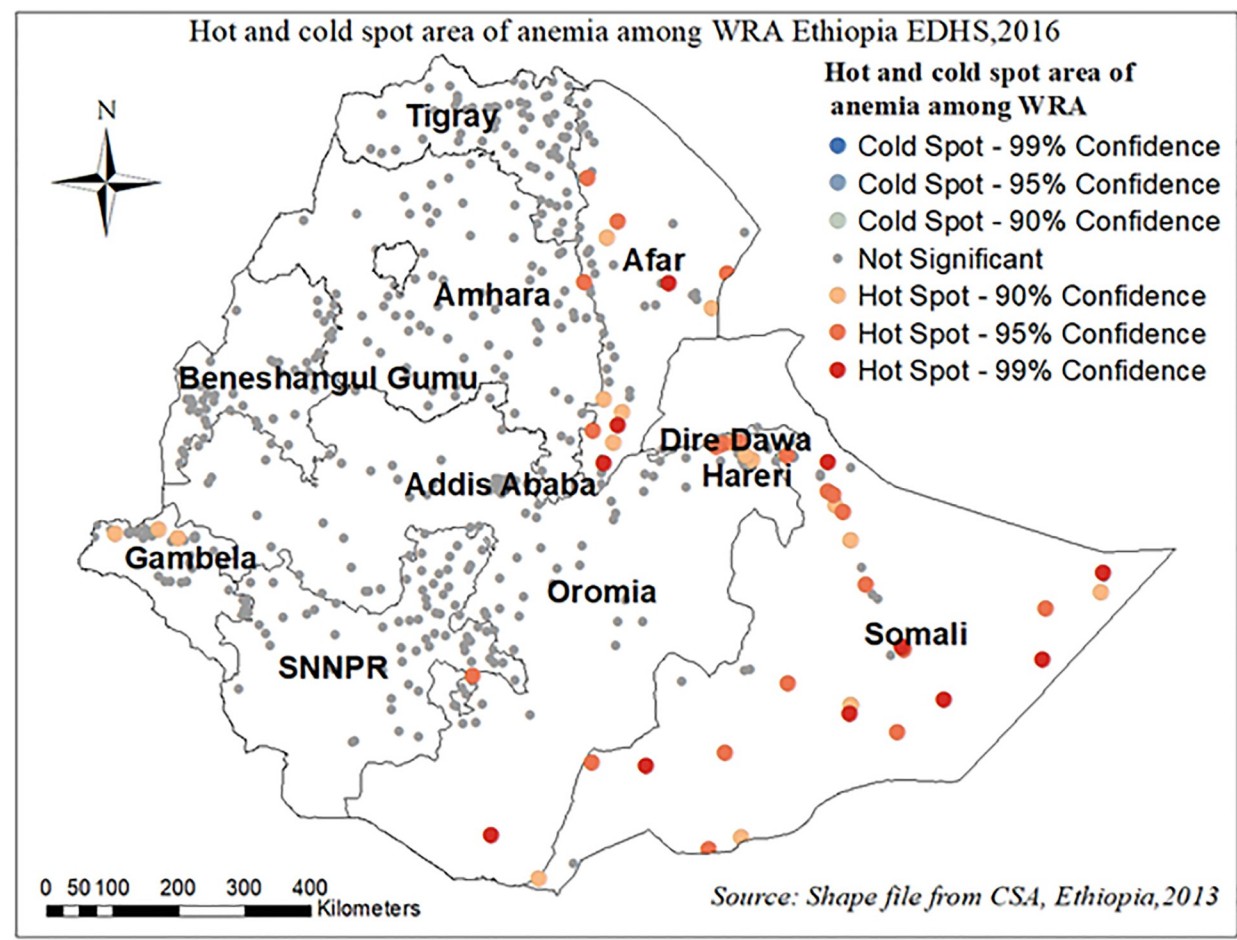

**Fig 7. Hot and cold spot area of anemia among reproductive age women in Ethiopia, EDHS 2016.**

**Table 4. Primary and Secondary SaTScan analysis result of anemia among reproductive-age women in Ethiopia Ethiopian demographic and health survey 2016.**

| Cluster | Enumeration area identified | Coordinate/radius | Population | cases | RR | LLR | P-value |
|---|---|---|---|---|---|---|---|
| 1(50) | 146, 138, 92, 490, 543, 492, 85, 358, 164, 77, 171, 198, 629, 95, 497, 278, 521, 588, 458, 553, 269, 318, 378, 187, 630, 214, 251, 573, 556, 239, 116, 22, 520, 33, 568, 277, 480, 527, 208, 64, 439, 57, 8, 210, 186, 394, 454, 436, 566, 212 | 6.023458 N, 44.807507 E / 462.80 km | 940 | 534 | 2.33 | 206.7 | P<0.0001 |
| 2(43) | 138, 164, 85, 358, 146, 492, 92, 490, 543, 278, 171, 198, 95, 318, 77, 187, 497, 556, 520, 629, 521, 588, 553, 458, 480, 208, 214, 251, 573, 239, 269, 116, 22, 394, 378, 630, 568, 33, 277, 286, 527, 289, 64 | 5.589269 N, 44.175032 E / 443.13 km | 810 | 472 | 2.37 | 192.8 | P<0.0001 |
| 3(30) | 366, 4, 427, 632, 440, 75, 596, 178, 499, 205, 334, 570, 599, 348, 544, 389, 241, 344, 332, 172, 571, 488, 191, 130, 249, 368, 189, 511, 55, 585 | 12.401068 N, 42.163134 E / 264.82 km | 566 | 269 | 1.85 | 59.4 | P<0.0001 |
| 4(23) | 307, 642, 1, 281, 242, 566, 523, 622, 288, 419, 381, 357, 311, 495, 610, 329, 352, 473, 202, 514, 493, 212, 238 | 9.748678 N, 42.299612 E / 49.24 km | 459 | 211 | 1.77 | 41.1 | P<0.0001 |
| 5(14) | 336, 39, 135, 102, 37, 564, 283, 484, 295, 620, 230, 51, 637, 491 | 9.963904 N, 40.440496 E / 81.80 km | 273 | 140 | 1.97 | 38.4 | P<0.0001 |
| 6(10) | 601, 82, 7, 398, 21, 50, 377, 422, 316, 182 | 4.211065 N, 38.646702 E / 202.47 km | 240 | 114 | 1.81 | 24.4 | P<0.0001 |
| 7(4) | 441, 557, 594, 30 | 9.488460 N, 41.736698 E / 11.91 km | 89 | 51 | 2.17 | 18.73 | P<0.0001 |
| 8(23) | 266, 618, 309, 435, 536, 370, 507, 592, 104, 260, 233, 69, 426, 603, 346, 315, 567, 343, 13, 105, 106, 417, 284 | 8.389747 N, 33.258557 E / 138.81 km | 398 | 147 | 1.41 | 10.5 | P = 0.019 |
| 9(1) | 445 | 6.376026 N, 38.396332 E / 0 km | 22 | 16 | 2.27 | 10.18 | P = 0.025 |

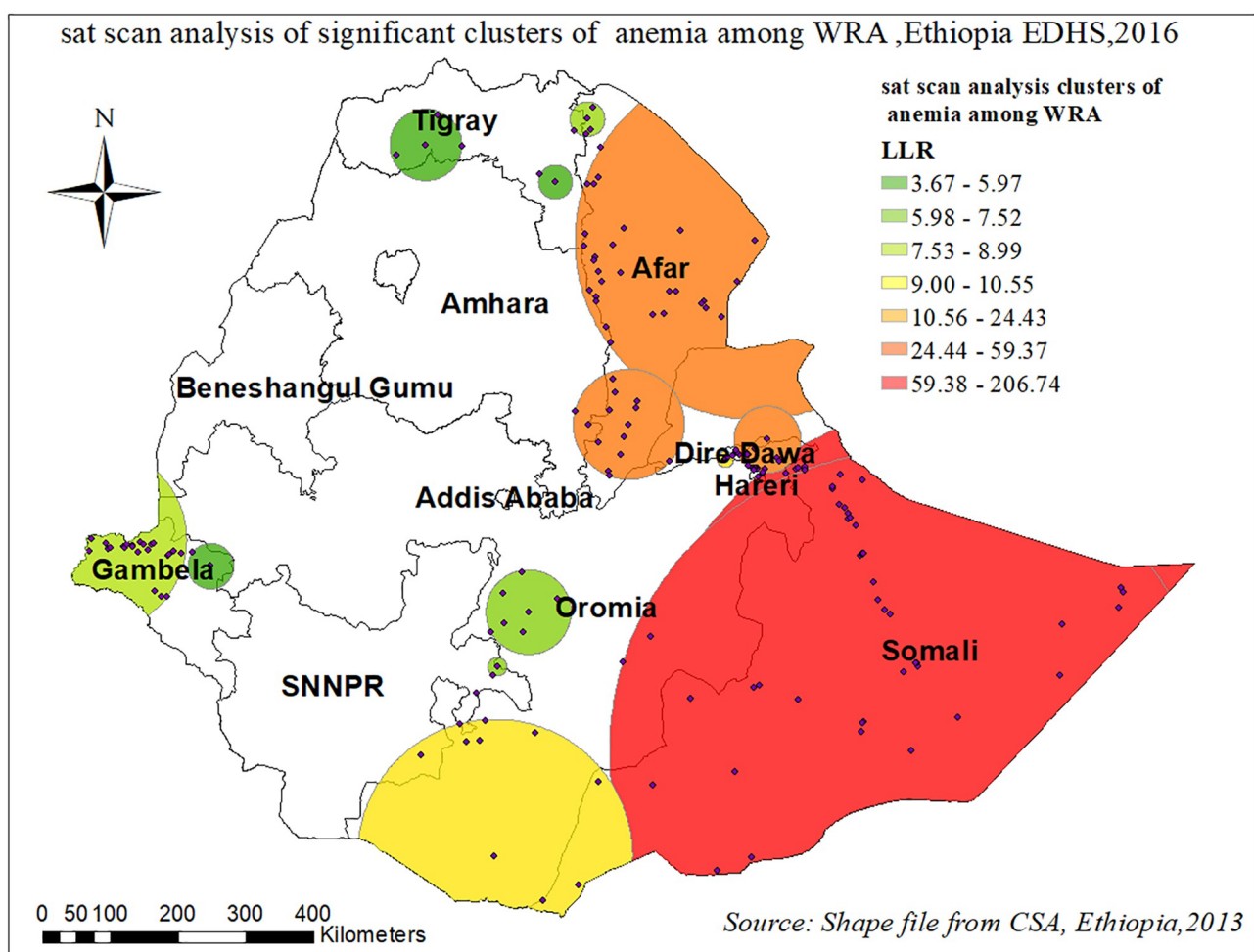

**Fig 8. Sat scan analysis of significant clusters of anemia among reproductive age women in Ethiopia, EDHS 2016.**

Reproductive age women who had a family size greater than five have higher odds of anemia as compared to having a family size less than two. This finding is in line with different studies [6, 9]. This could be to the possibility that a large family size leads to food insecurity in the home, jeopardizing women's access to a healthy diet.

In this study reproductive-age women who were married have more likely to have anemia as compared to all other marital statuses. This is in line with a study conducted in Ethiopia [28], East Africa [22], but different from the 2005 EDHS report [29], and Ruanda [30]. This might be most married women become pregnant and lactating, and the complications that result may make them anemic.

In this study reproductive age women who use pills, injectable or implant contraceptives were less likely to be affected by anemia. This is in line with the study in East Africa, Ruanda, Nepal [22, 25, 30]. This is because women who used this type of contraceptive method with high efficiency to prevent pregnancy result in complications related to pregnancy and childbirth.

By themself hormonal contraceptive methods could minimize menstrual bleeding, besides, the noncontraceptive iron content pills are also used for the prevention of heavy menstrual bleeding and regulating menses [6].

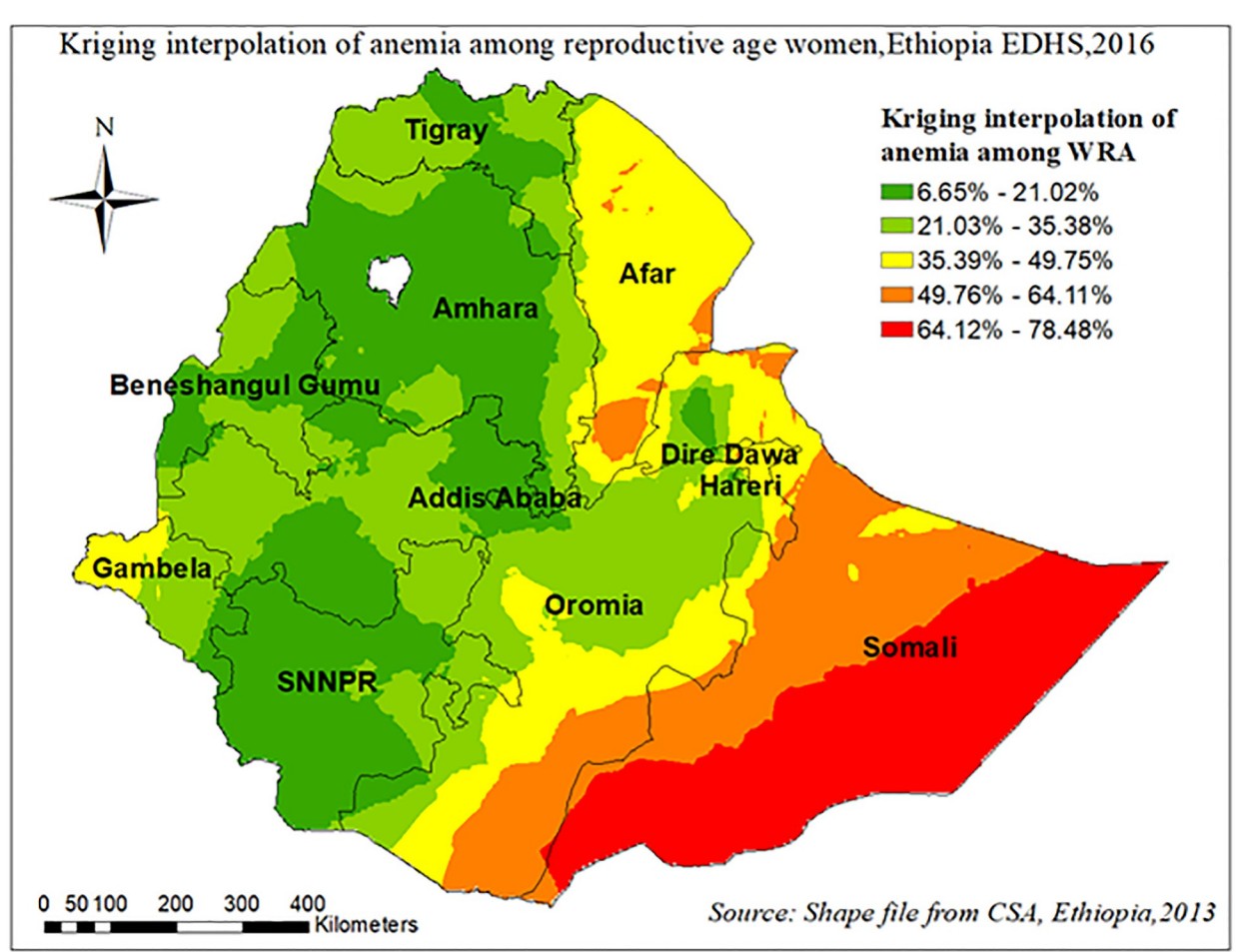

**Fig 9. Kriging interpolation of anemia among reproductive age women in Ethiopia, EDHS 2016.**

**Table 5. Summary of OLS results and diagnostics for anemia among WRA in Ethiopia, EDHS 2016.**

| Variable | Coefficients | Standard error | t-statistics | Probability | Robust standard error | Robust statistic | Robust probability | VIF |
|---|---|---|---|---|---|---|---|---|
| Intercept | 0.406 | 0.037 | 10.73 | <0.001 | 0.038 | 10.50 | <0.001 | - - - |
| Women have rich wealth status | -0.033 | .0255 | -1.30 | 0.19 | 0.025 | -1.312 | 0.189 | 2.76 |
| Women have more than 5 family members | -0.029 | 0.038 | -0.77 | 0.44 | 0.036 | -0.81 | 0.42 | 1.46 |
| Women who have formal education | -0.125 | 0.035 | -3.51 | <0.001 | 0.035 | -3.53 | <0.001 | 2.73 |
| Women who use Pills/injectable/implant | -0.392 | 0.046 | -8.58 | <0.001 | 0.045 | -8.65 | <0.001 | 1.34 |
| Women have more than 1 child within five years | 0.233 | 0.055 | 4.25 | <0.001 | 0.061 | 3.78 | 0.002 | 2.16 |
| **OLS diagnostics** | | | | | | | | |
| Number of observation | | 621 | | Akaike's Information Criterion (AIC) | | | -547.28 | |
| Multiple R-squared | | 0.36 | | Adjusted R-Squared | | | 0.355 | |
| Joint F-statistic: | | 69.42 | | Prob(> F), (5615) degrees of freedom | | | <0.001 | |
| Joint Wald Statistic | | 303.92 | | Prob(> chi-squared), (5) degrees of freedom | | | <0.001 | |
| Koenker (BP) Statistics | | 56.69 | | Prob(> chi-squared), (5) degrees of freedom | | | <0.001 | |
| Jarque-Bera Statistics: | | 13.11 | | Prob(> chi-squared), (2) degrees of freedom | | | 0.0014 | |

**Table 6. Geographically weighted regression (GWR) model for anemia among WRA delivery in Ethiopia, EDHS 2016.**

| Explanatory variable | Women who have formal education, women who use Pills/injectable/implant, women who have more than 1 child within five years, women who have rich wealth status, and women who have more than 5 family members |
| --- | --- |
| Residual squares | 9.45 |
| Effective number | 82.89 |
| Sigma | 0.13 |
| Akaike's Information Criterion (AICc) | -697 |
| Multiple R-Squared | 0.59 |
| Adjusted R-Squared | 0.53 |

The odds of having anemia among reproductive age women who gave birth to more than one child within five years were higher than not having birth within a specified period. This finding is in line with a study in Ethiopia [9], India [8] might be that narrow birth interval delays the restoration of iron and other micronutrient stores in the body between pregnancies, and also women with frequent birth history could have a history of obstetric complications such as postpartum hemorrhage and sepsis which directly expose them to anemia [1, 31].

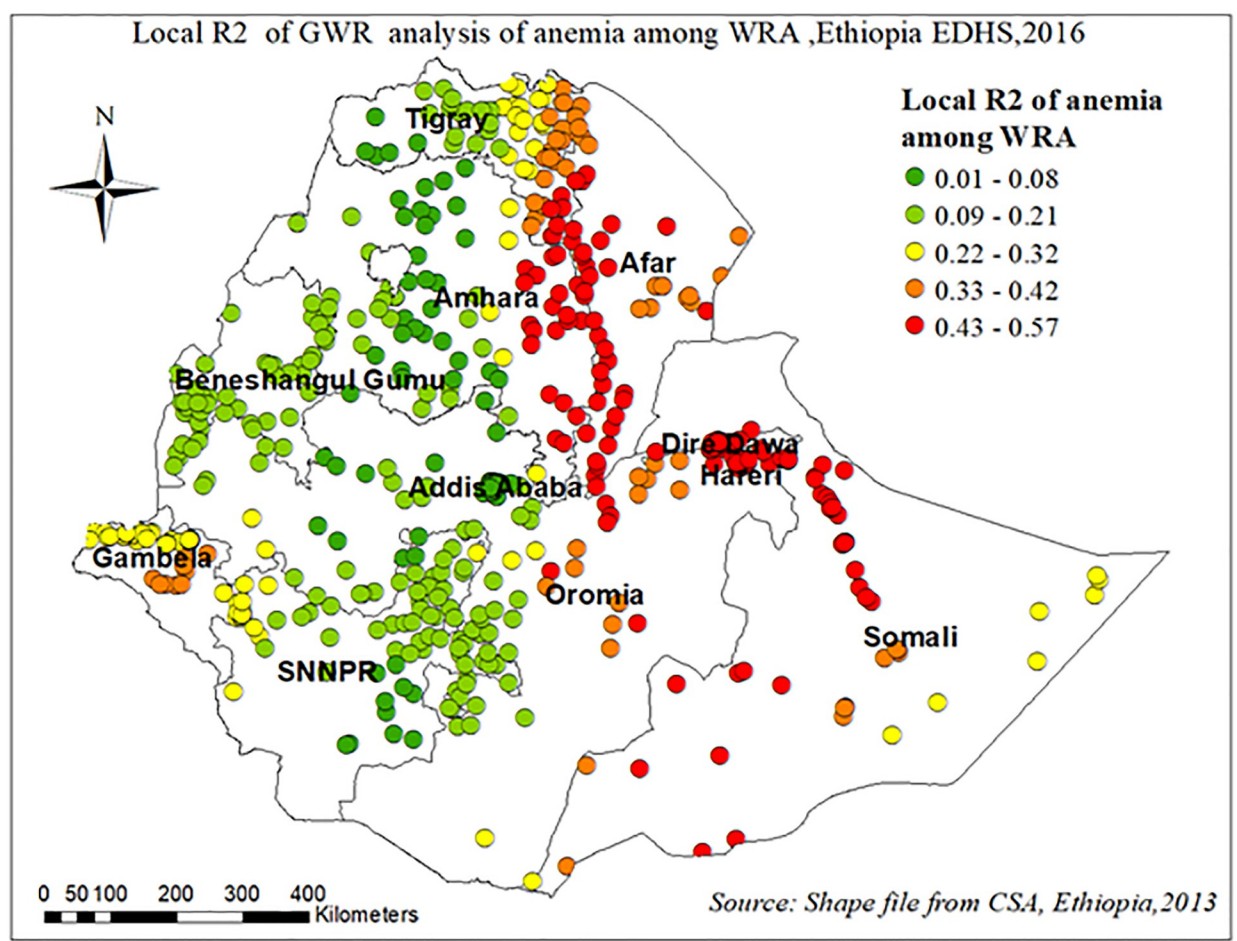

**Fig 10. Local R2 of GWR analysis anemia among reproductive age women in Ethiopia, EDHS 2016.**

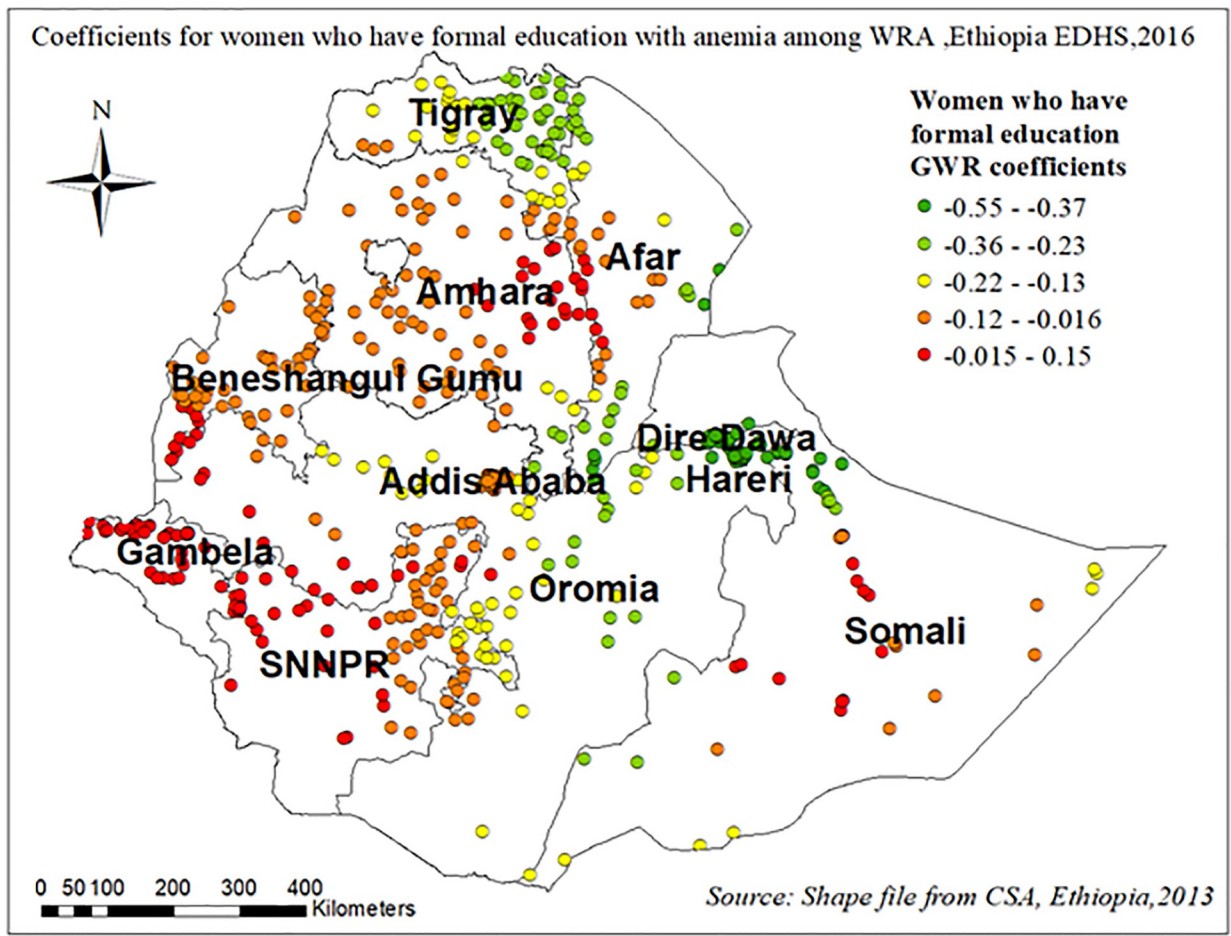

**Fig 11. Coefficients for women who have formal education with anemia among reproductive-age women in Ethiopia, EDHS 2016.**

In this study, reproductive age women who were rich are less likely to have anemia when compared to poor and poorest women. This is in line with different studies [15, 22, 30, 32, 33]. This could be due to that when a woman has improved her wealth status, she enables to purchase healthy nutrition and can utilize health services [33].

Reproductive age women who were attained primary and more than primary education were less likely to be affected by anemia. This is in line with a study in Ethiopia [6], Rwanda [30]. This might be that women who have education have higher health-seeking behavior and service utilization than non educated women so that they couldn't get preventive and curative services for conditions that contribute to anemia.

The finding of this study showed that anemia among reproductive age women had significant spatial variation in Ethiopia. The spatial SaTscan statistics detected a total of 198 significant clusters with a high prevalence of anemia among reproductive age women. The Somali, Dire Dewa, and Afar regions of Ethiopia were shown to have significant hotspot areas of anemia among WRA. Whereas, Amhara, SNNPR, and Tigray regions of Ethiopia were less risk areas. Studies conducted in Ethiopia [7, 34, 35] and other developing countries [36, 37] also pointed out the significant regional variations in the use of anemia among WRA. Moreover, the multilevel result revealed that the odds of having anemia among reproductive age women who were living in Somalia and the Afar region were higher as compared to the Tigray region.

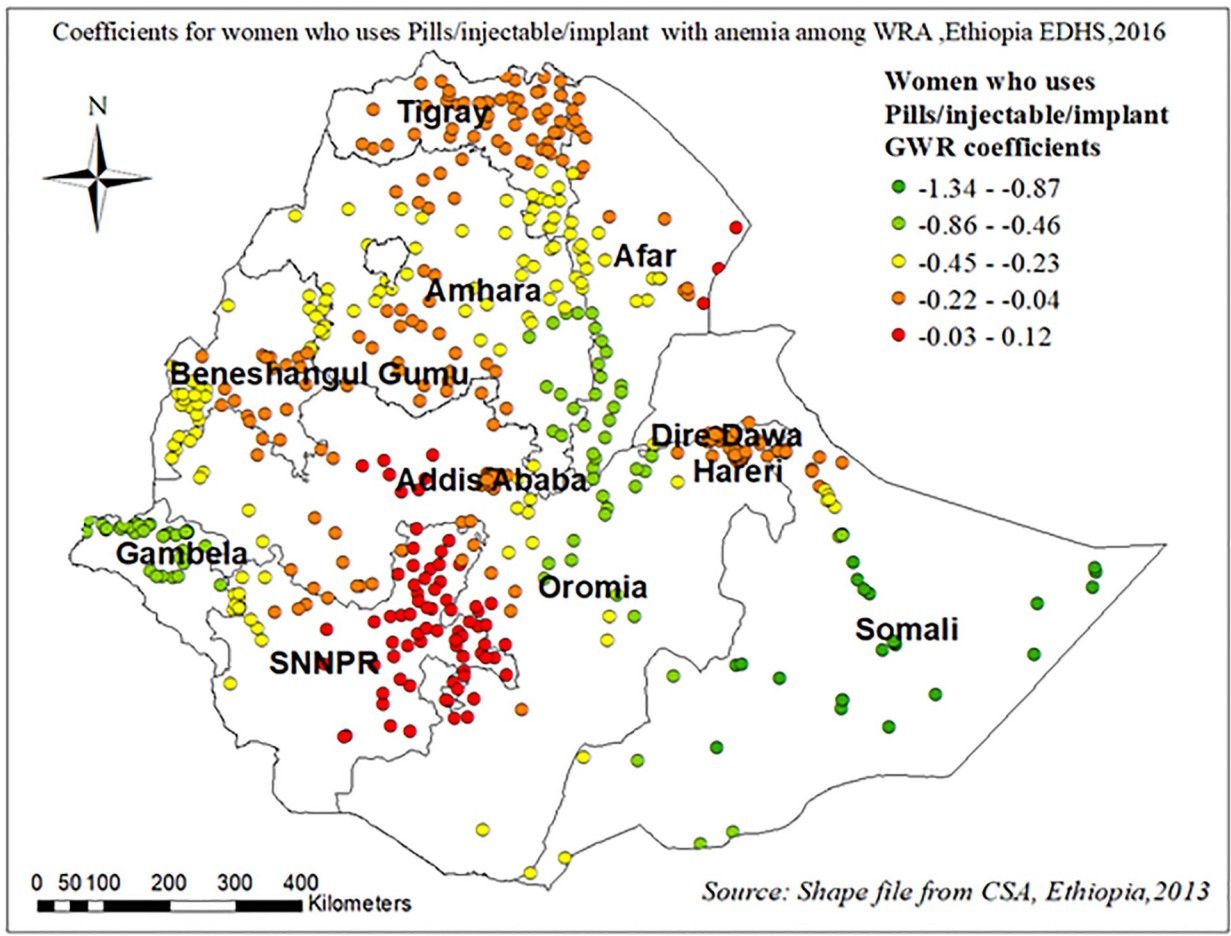

**Fig 12. Coefficients for women who use pills, injectable contraceptives, and implants with anemia among reproductive-age women in Ethiopia, EDHS 2016.**

This is in line with different studies conducted in Ethiopia [7, 34], Tanzania [36]. This might result from differences in dietary preferences and disease burden, inequalities in access to health care across the regions and differences in societal beliefs, cultural practices towards the care for women. and recurrent drought may be triggered food insecurity might have contributed to the higher prevalence of anemia in these regions [6].

The GWR analysis revealed that there is a negative relationship between women having formal education and women who use Pills/injectables/implants with anemia among WRA. However, women who had more than 1 child within five years were more likely to have anemia in multiple regions of Ethiopia. The findings from the GWR analysis were similar to the multilevel analysis conducted in this study.

The utilization of nationally representative data with a high sample size was the study's key strength. Another advantage was that performing multilevel analysis to adjust for the data's correlated nature. The use of spatial analysis including modeling spatial relationships using GWR was also another strength of this study which was used to identify factors that contributed to spatial variation of anemia among WRA. This research, however, has certain flaws. We weren't able to include crucial elements such as hookworm infestation and diet type since we used secondary data.

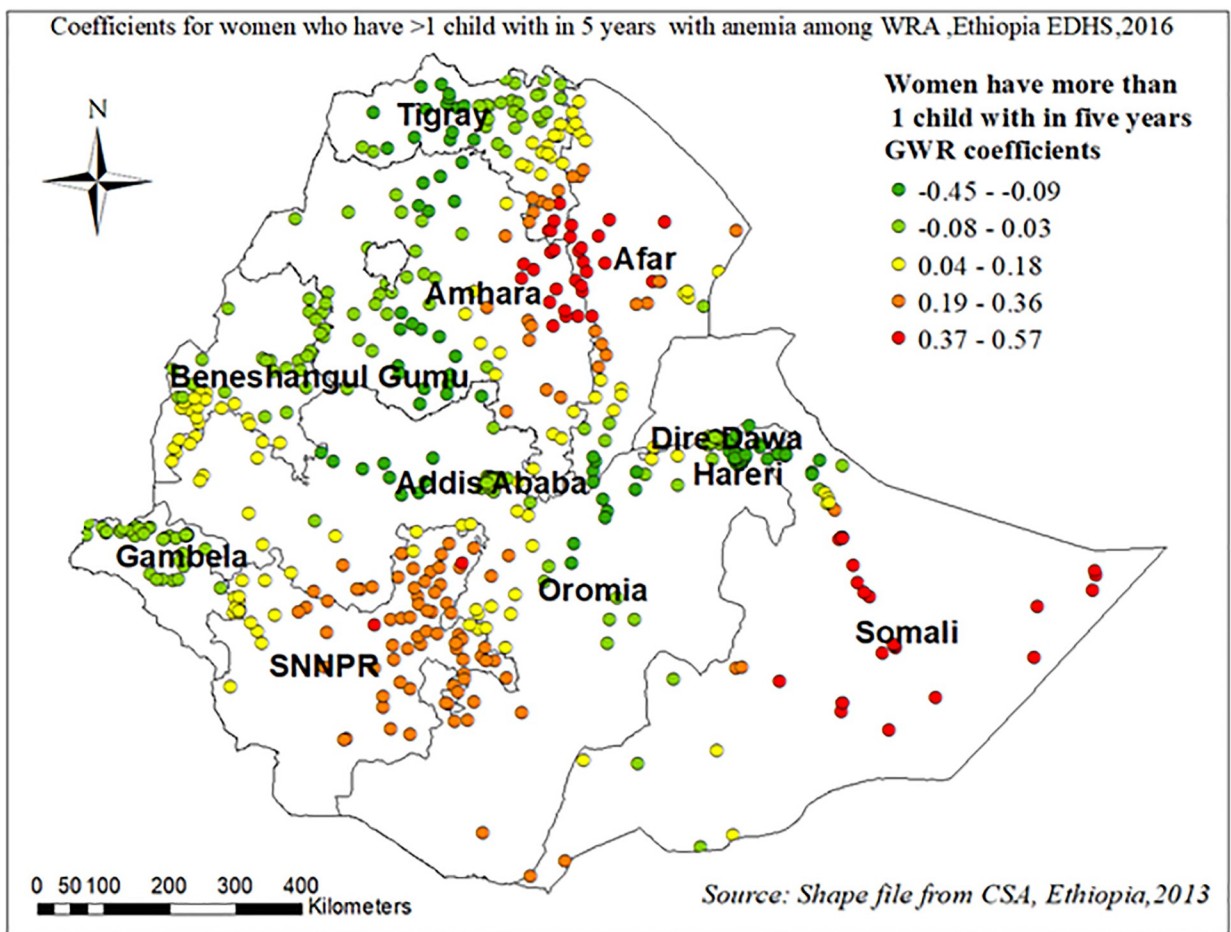

**Fig 13. Coefficients for women who have more than 1 child within five years with anemia among reproductive age women in Ethiopia, EDHS 2016.**

## Conclusion

In Ethiopia, anemia among reproductive-age women had spatial variations across the regions. The GWR analysis shows that mothers having a formal education, and women who use Pills/injectable/implant decreases the risks of anemia among reproductive-age women. However, women who have more than one child within five years increased the risk of anemia among reproductive-age women in Ethiopia. Therefore, it is important that the Ethiopia FMoH pay special attention to those groups of women who have a higher prevalence of anemia, such as an increased number of births, being married, and increase family size is recommended. Women's education and family planning usage especially pills, implants, or injectable should be strengthened. Anemia prevention and control programmers should be a strength for WRA living in high anemic areas such as in Afar, Somali and Dire Dawa regions.

## Supporting information

**S1 Fig.**
(TIF)

**S1 File.**
(DOCX)

## Acknowledgments

We would like to acknowledge the Measure DHS program which permitted us to use DHS data. We would also thank the Central Statistical Agency for providing the shapefile.

## Author Contributions

**Conceptualization:** Daniel Gashaneh Belay, Shumet Mebrat Adane, Oshe Lemita Ferede, Ayenew Molla Lakew.

**Data curation:** Daniel Gashaneh Belay, Shumet Mebrat Adane, Oshe Lemita Ferede.

**Formal analysis:** Daniel Gashaneh Belay, Shumet Mebrat Adane, Oshe Lemita Ferede.

**Funding acquisition:** Daniel Gashaneh Belay, Shumet Mebrat Adane.

**Investigation:** Daniel Gashaneh Belay, Shumet Mebrat Adane, Oshe Lemita Ferede.

**Methodology:** Daniel Gashaneh Belay, Oshe Lemita Ferede.

**Project administration:** Daniel Gashaneh Belay, Shumet Mebrat Adane, Oshe Lemita Ferede.

**Resources:** Daniel Gashaneh Belay, Shumet Mebrat Adane, Oshe Lemita Ferede.

**Software:** Daniel Gashaneh Belay, Shumet Mebrat Adane, Oshe Lemita Ferede, Ayenew Molla Lakew.

**Supervision:** Daniel Gashaneh Belay, Shumet Mebrat Adane, Oshe Lemita Ferede, Ayenew Molla Lakew.

**Validation:** Daniel Gashaneh Belay, Shumet Mebrat Adane, Oshe Lemita Ferede, Ayenew Molla Lakew.

**Visualization:** Daniel Gashaneh Belay, Shumet Mebrat Adane, Oshe Lemita Ferede, Ayenew Molla Lakew.

**Writing – original draft:** Daniel Gashaneh Belay, Shumet Mebrat Adane, Oshe Lemita Ferede.

**Writing – review & editing:** Daniel Gashaneh Belay, Shumet Mebrat Adane, Ayenew Molla Lakew.

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
