## [Decision Letter · Decision Letter 0]

9 Jun 2021

PONE-D-20-35910

Geographically weighted regression analysis of anemia and its associated factors among reproductive age women in Ethiopia using the 2016 Demographic and Health Survey.

PLOS ONE

Dear Dr. Belay,

Thank you for submitting your manuscript to PLOS ONE. After careful consideration, we feel that it has merit but does not fully meet PLOS ONE’s publication criteria as it currently stands. Therefore, we invite you to submit a revised version of the manuscript that addresses the points raised during the review process.

We look forward to receiving your revised manuscript.

Kind regards,

Lucy C. Okell

Academic Editor

PLOS ONE

2.Thank you for submitting the above manuscript to PLOS ONE. During our internal evaluation of the manuscript, we found significant text overlap between your submission and the following previously published works:

 -https://www.researchsquare.com/article/rs-8809/v1 ("Spatial patterns and associated factors’ of Early Marriage among Reproductive age women in Ethiopia: a Secondary Analysis of EDHS 2016" by Zemenu Tessema Tadesse; please also be advised that this is a preprint which has not undergone peer review and should not be interpreted as an endorsement of its validity or suitability for dissemination as established information or for guiding clinical practice)

- https://onlinelibrary.wiley.com/doi/full/10.1111/mcn.13063 ("Factors associated with anaemia among women of reproductive age in Ethiopia: Multilevel ordinal logistic regression analysis" by Tirore et al.)

- https://journals.plos.org/plosone/article?id=10.1371%2Fjournal.pone.0236449 ("Prevalence and factors associated with anemia among women of reproductive age in seven South and Southeast Asian countries: Evidence from nationally representative surveys" by Sunuwar et al.)

- https://pubmed.ncbi.nlm.nih.gov/32760116/ ("Spatiotemporal patterns of anemia among lactating mothers in Ethiopia using data from Ethiopian Demographic and Health Surveys (2005, 2011 and 2016)" by Liyew et al.)

- https://bmcpublichealth.biomedcentral.com/articles/10.1186/s12889-020-08934-9 ("Individual and community level factors associated with anemia among lactating mothers in Ethiopia using data from Ethiopian demographic and health survey, 2016; a multilevel analysis" by Liyew and Teshale)

We would like to make you aware that copying extracts from previous publications word-for-word is unacceptable. In addition, the reproduction of text from published reports has implications for the copyright that may apply to the publications.

Please revise the manuscript to rephrase the duplicated text, cite your sources, and provide details as to how the current manuscript advances on previous work. Please note that further consideration is dependent on the submission of a manuscript that addresses these concerns about the overlap in text with published work.

We will carefully review your manuscript upon resubmission, so please ensure that your revision is thorough."

3. In your ethics statement in the Methods section and in the online submission form, please provide additional information about the data used in your retrospective study. Specifically, please ensure that you have discussed whether all data were fully anonymized before you accessed them and/or whether the IRB or ethics committee waived the requirement for informed consent. If patients provided informed written consent to have data from their medical records used in research, please include this information.

5. We note that Figure 4,7,8,9,10,11,12 & 13 in your submission contain map images which may be copyrighted. All PLOS content is published under the Creative Commons Attribution License (CC BY 4.0), which means that the manuscript, images, and Supporting Information files will be freely available online, and any third party is permitted to access, download, copy, distribute, and use these materials in any way, even commercially, with proper attribution. For these reasons, we cannot publish previously copyrighted maps or satellite images created using proprietary data, such as Google software (Google Maps, Street View, and Earth). For more information, see our copyright guidelines: http://journals.plos.org/plosone/s/licenses-and-copyright.

a. You may seek permission from the original copyright holder of Figure  4,7,8,9,10,11,12 & 13  to publish the content specifically under the CC BY 4.0 license. 

2. Thank you for submitting the above manuscript to PLOS ONE. During our internal evaluation of the manuscript, we found significant text overlap between your submission and the following previously published works:

- https://www.researchsquare.com/article/rs-8809/v1 ("Spatial patterns and associated factors’ of Early Marriage among Reproductive age women in Ethiopia: a Secondary Analysis of EDHS 2016" by Zemenu Tessema Tadesse; please also be advised that this is a preprint which has not undergone peer review and should not be interpreted as an endorsement of its validity or suitability for dissemination as established information or for guiding clinical practice)

- https://onlinelibrary.wiley.com/doi/full/10.1111/mcn.13063 ("Factors associated with anaemia among women of reproductive age in Ethiopia: Multilevel ordinal logistic regression analysis" by Tirore et al.)

- https://journals.plos.org/plosone/article?id=10.1371%2Fjournal.pone.0236449 ("Prevalence and factors associated with anemia among women of reproductive age in seven South and Southeast Asian countries: Evidence from nationally representative surveys" by Sunuwar et al.)

- https://pubmed.ncbi.nlm.nih.gov/32760116/ ("Spatiotemporal patterns of anemia among lactating mothers in Ethiopia using data from Ethiopian Demographic and Health Surveys (2005, 2011 and 2016)" by Liyew et al.)

- https://bmcpublichealth.biomedcentral.com/articles/10.1186/s12889-020-08934-9 ("Individual and community level factors associated with anemia among lactating mothers in Ethiopia using data from Ethiopian demographic and health survey, 2016; a multilevel analysis" by Liyew and Teshale)

We would like to make you aware that copying extracts from previous publications word-for-word is unacceptable. In addition, the reproduction of text from published reports has implications for the copyright that may apply to the publications.

Please revise the manuscript to rephrase the duplicated text, cite your sources, and provide details as to how the current manuscript advances on previous work. Please note that further consideration is dependent on the submission of a manuscript that addresses these concerns about the overlap in text with published work.

3. In your ethics statement in the Methods section and in the online submission form, please provide additional information about the data used in your retrospective study. Specifically, please ensure that you have discussed whether all data were fully anonymized before you accessed them and/or whether the IRB or ethics committee waived the requirement for informed consent. If patients provided informed written consent to have data from their medical records used in research, please include this information.

4. We note that Figures 4, 5, 7-13 in your submission contain map images which may be copyrighted. All PLOS content is published under the Creative Commons Attribution License (CC BY 4.0), which means that the manuscript, images, and Supporting Information files will be freely available online, and any third party is permitted to access, download, copy, distribute, and use these materials in any way, even commercially, with proper attribution. For these reasons, we cannot publish previously copyrighted maps or satellite images created using proprietary data, such as Google software (Google Maps, Street View, and Earth). For more information, see our copyright guidelines: http://journals.plos.org/plosone/s/licenses-and-copyright.

4.1.    You may seek permission from the original copyright holder of Figures 4, 5, 7-13 to publish the content specifically under the CC BY 4.0 license. 

4.2.    If you are unable to obtain permission from the original copyright holder to publish these figures under the CC BY 4.0 license or if the copyright holder’s requirements are incompatible with the CC BY 4.0 license, please either i) remove the figure or ii) supply a replacement figure that complies with the CC BY 4.0 license. Please check copyright information on all replacement figures and update the figure caption with source information. If applicable, please specify in the figure caption text when a figure is similar but not identical to the original image and is therefore for illustrative purposes only.

Reviewers' comments:

Reviewer's Responses to Questions

**Comments to the Author**

1. Is the manuscript technically sound, and do the data support the conclusions?

Reviewer #1: Yes

Reviewer #2: No

2. Has the statistical analysis been performed appropriately and rigorously? 

Reviewer #1: Yes

Reviewer #2: No

3. Have the authors made all data underlying the findings in their manuscript fully available?

Reviewer #1: Yes

Reviewer #2: Yes

4. Is the manuscript presented in an intelligible fashion and written in standard English?

Reviewer #1: Yes

Reviewer #2: No

5. Review Comments to the Author

Reviewer #1: This is a rather comprehensive mathematical and statistical approach to the data analysis using the fixed effects (a measure of association), random-effects (a measure of variation), spatial autocorrelation and hot spot analysis with spatial interpolation as well as spatial scan. Also, ordinary least squares approaches were applied as well as weighted least squares in the geographical interpretation of results.

The study used population-based cross-sectional survey data from the 2016 Demographic Health Surveys conducted in Ethiopia. Ethiopia. The source population was all women aged 15 to 49 within five years before the survey in Ethiopia, while all reproductive-age women in the selected enumeration areas were the study population. The sample was certainly adequate and consisted of women whose hemoglobin level was done (14,489 )and of them about 14,171were included in the study.

According to the investigators this is nationally representative data which In Ethiopia, anemia among reproductive-age women had spatial variations across the regions. Interestingly, the cluster patterns show high rates of anemia among reproductive age women occurrence over the study area. the Z-score of 23.35 indicated that there is less than 1% likelihood that this cluster pattern could result from random chance as seen from Figure 5.

Although the mathematical development was, in fact, comprehensive, the results are basically descriptive. The plots and graphs were well formatted and explanatory.

On a minor note, the English needs editing for proper English construction and grammar.

Reviewer #2: Abstract

The defininition of anaemia is not clearly defined. I suggest the authors to specificy the actual threshold in g/dL as is defined by the WHO or the Ethiopian Ministry of Health.

The conclusion should clearly speak more on the policy imperatives amidst the disparities highlighted.

Introduction

The introduction should re-written to capture the global, SSA and national dynamics of anaemia in Ethiopia. This should be followed by the epidemiological significance of anaemia in Epthiopia.

The clinical defininition of anaemia is not concise. I suggest the authors to specificy the actual threshold in g/dL as is defined by the WHO or the Ethiopian Ministry of Health. Less than normal is not quite clear.

The full definition of WRA is not outlined when it is first mentioned.

I suggest an English editor.

Materials and methods

The authors indicate that a fixed effects were used to estimate the association between the likelihood of anaemia. They futher indicate that the random effects were used to estimate the median odds ratio.

We expect the authors to detail the regression model and outline how the fixed and random effects are to be conceptualized.

Some terms used in the defining the model parameters are not defined e,g ICC, VA.

A supplemntary file of the modelling framework will be useful.

I am not sure if ordinary least square analysis as is expreessed is appropriate for the methods section.

Results

This section should be rewritten and the model output discussed as per the methodology previously employed.

Whereas there are many factors associated with anaemia from the multi-level analysis in Table 3. The authors have only discussed the age of the women and how it relates to ananemia. Futhermore the wording and odds-ration interpretation is not clear and need to be looked at possibly with the help of a bio-statistician.

The seems to be too many figures in the manuscripts (Figure 1 – Figure 13). I suggest that some figures could be added as a supplementary.

Discussion

We need to know more on how the study findings relate to other settings in SSA as compared to Ethiopia.

6. PLOS authors have the option to publish the peer review history of their article (what does this mean?). If published, this will include your full peer review and any attached files.

Reviewer #1: No

Reviewer #2: **Yes: **Julius Nyerere Odhiambo

---

## [Author Response · Author response to Decision Letter 0]

2 Oct 2021

Point by point response to reviewers comment 

Manuscript title: Geographically weighted regression analysis of anemia and its associated factors among reproductive age women in Ethiopia using the 2016 Demographic and Health Survey.

Manuscript number: PONE-D-20-35910 

Dear editor and reviewers, thank you for the most important issues you raised for the betterment of our manuscript. Below are the point-by-point response for the comments and concerns you raised. In addition, we have incorporated the comments and concerns in the revised manuscript. 

 Response to reviewer’s comments 

 Response to Reviewer #1

1. The English needs editing for proper English construction and grammar.

Authors’ response: Thank you for your comment. We have taken the comment and we amend the manuscript for grammatical errors, coherency, and consistency. 

Response to reviewer #2

Abstract

1. The definition of anemia is not clearly defined, the actual threshold in g/dL as is defined by the WHO or the Ethiopian Ministry of Health.

Authors’ response: Thank you for your important comment. We have corrected as,

Anemia is defined as the hemoglobin level <110 g/dl for lactating or pregnant mothers and hemoglobin level <120 g/dl for none pregnant or non-lactating women in abstract and introduction section in line no. 23-24 and 51-52.

2. The conclusion should clearly speak more on the policy imperatives amidst the disparities highlighted.

Authors’ response: Thank you for your important comment. We have amended it accordingly in abstract section page 2 in line no. 43-46.

Introduction

3. The introduction should re-write to capture the global, SSA and national dynamics of anemia in Ethiopia. This should be followed by the epidemiological significance of anemia in Ethiopia.

Authors’ response: Thank you for raising this important comment. We have taken the comment and corrected made in the revised version of the manuscript in introduction section page 3 in line no. 67-75.

4. The clinical definition of anemia is not concise. Less than normal is not quite clear.

Authors’ response: Thank you for your comment. We have taken the comment and amendment made in the revised version of the manuscript. Anemia is defined as the hemoglobin level <110 g/dl for lactating or pregnant mothers and hemoglobin level <120 g/dl for none pregnant or non-lactating women. We corrected on page 2&3 in abstract and introduction section in line no. 23-24 and 51-52.

5. The full definition of WRA is not outlined when it is first mentioned.

Authors’ response: Thank you for raising this important comment. We have taken the comment and amendment made in the revised version of the manuscript by saying Women at Reproductive Age (WRA).it is amended in page 4 and line no. 85.

 Methods

6. The authors indicate that fixed effects were used to estimate the association between the likelihood of anemia. They further indicate that the random effects were used to estimate the median odds ratio.

We expect the authors to detail the regression model and outline how the fixed and random effects are to be conceptualized.

Authors’ response: Thank you for raising this important comment. 

In our study since the ICC was greater than 5% and the MOR was significant, doing multilevel analysis was necessary. Mixed effect which means both fixed effect and random effect mode parameter were used for parameter estimation for multilevel analysis. The fixed effects (a measure of association) were used to estimate the association between the likelihood of anemia among women at reproductive age and explanatory variables which was reported by adjusted odds ratio (AOR) with 95% CI.. Whereas, random-effects (a measure of variation) is used to estimate the variation of anemia prevalence among reproductive age between clusters which assessed by Median odds ratio (MOR), by intraclass correlation coefficient (ICC), and Proportional Change in Variance (PCV). Specifically, the MOR can be understood as the increased risk (in median) that would have if a women move to another cluster from the low risk to a higher risk clusters (cluster level difference of odds ratio). It is corrected on page 7 and line no 168-193.

7. Some terms used in the defining the model parameters are not defined e,g ICC, VA.

Authors’ response: Thank you for raising this comment. We have taken the comment and amendment made in the revised version of the manuscript. It is corrected on page 7 and line no 168-193.

8. A supplementary file of the modeling framework will be useful.

Authors’ response: Thank you for raising this comment. We have taken the comment and prepared a modeling frame work of anemia among WRA in Ethiopia and added in supplementary file (S1) of the revised version of the manuscript.

9. I am not sure if ordinary least square analysis as is expressed is appropriate for the methods section.

Authors’ response: Thank you for raising this comment. To assess factors which affecting the spatial variation of anemia among reproductive-age women both ordinary least squares (OLS) analysis and geographical weighted regression are needed. The parameters results in OLS such as Joint Wald statistic, multicollinearity, and Moran’s I value are necessary to perform the geographical weighted regression.

 Results

10. This section should be rewritten and the model output discussed as per the methodology previously employed.

Authors’ response: Thank you for raising this comment. We have taken the comment and amendment made in the revised version of the manuscript by rewriting the result in statistical way in age 14 line no. 284-295.

11. There are many factors associated with anemia from the multi-level analysis in Table 3. The authors have only discussed the age of the women and how it relates to anemia. 

Authors’ response: Thank you for raising this comment. Dear review it is known that we are doing a research in women at reproductive age (WRA) in Ethiopia, therefore WRA were our source of population that we are generalized on them. But age was not significant variables for anemia among WRA rather variables such as marital status of women, education status of women, wealth index of the household, family size, hormonal contraceptive usage, and region they live had significant association with anemia among reproductive age women. Then we are discussed them in revised manuscript in detail starting from the prevalence of Anemia among WRA to all significant variables. See the detail in page 24-26 line no.431-489.

12. The seems to be too many figures in the manuscripts (Figure 1 – Figure 13). I suggest that some figures could be added as a supplementary.

Authors’ response: Thank you for raising this comment. Dear reviewer all the image gives equally important information for each section. We have added one supplementary files

13. We need to know more on how the study findings relate to other settings in SSA as compared to Ethiopia.

Authors’ response: Thank you for raising this comment. We have taken the comment and amendment made in the revised version of the manuscript by including a research done in sub-Saharan African countries. See the detail in page 24 line no.420-426.

---

## [Editor Report · Decision Letter 1]

5 Jan 2022

PONE-D-20-35910R1Geographically weighted regression analysis of anemia and its associated factors among reproductive age women in Ethiopia using the 2016 Demographic and Health SurveyPLOS ONE

Dear Dr. Belay,

Thank you for submitting your manuscript to PLOS ONE. After careful consideration, we feel that it has merit but does not fully meet PLOS ONE’s publication criteria as it currently stands. Therefore, we invite you to submit a revised version of the manuscript that addresses the points raised during the review process.

Please make the following editorial checks and changes:There is an error in the anemia units. The definition should be 120 g/L not g/dLPlease run a full spelling check on the manuscript. E.g. see typos in the abstract: “identifay” -> identify and main text “pridictors” -> predictors.Line 50 suggest rewording “Reproductive age women are women who are found in the age from 15 up to 49 full-term years”to:Reproductive age is commonly defined among women as ages 15 to 49 years.Line 57: suggest rewording: “It is one of the most serious dangers to children's health and a factor in maternal mortality, because ity increasesd the risk of adverse pregnancy outcomes, child mortality, impaired neurocognitive abilities, and physical development of children, and reducesd work capacity despite being straightforward to 60 preventotect and treat [1, 3].”Line 69: “the pooled prevalence of anemia in pregnant women was 31.66% in Ethiopia” Do you have the year for this metric?Line 70: replace “lactating women” with ‘current lactation’Line 73: small ‘a’ for “Anemia’ – please correct this throughout the text.Line 83: “supplying”? (not supplaying)Line 104: “de juries” please define.Line 118 “khat” please defineLine 126 “we recode them based on the appropriate measure of central tendency” This is unclear – what does it mean? Centralise / normalise based on mean and SD?Line 149: “used for parameter estimation” please change to ‘included in the model’.Line 150: “among reproductive age between clusters” Please change to “among reproductive age women between clusters.Line 151. This does not quite make sense as the metrics you are mentioning are not the same as random effects. Please reword “It was estimated by…” to “We estimated the…”Line 156: please change “that would have if” to ‘associated with’Line 163: I did not follow this sentence, I think it could be removed: “In this study, since the ICC was greater than 5% and the MOR have significant difference, doing the multilevel analysis was necessary”  as the methods are already clear.Line 200. Sentence seems incomplete: “Unlike OLS that fits a single linear regression equation to all of the data in the study area, GWR 200 creates an equation for each.” …. Each?Table 1: please indicate what the numbers in brackets are.Fig 3: Pia chart – do you mean Pie chart?Line 233: this sentence needs rewording, grammar is incorrect and meaning unclear: “The MOR value in the null model also showed that, between higher and lower odd clusters anemic among reproductive-age women were different by 2.28 times.”Line 236 replace “expressed” with ‘explained’.Line 286: “from these 50 of these”  please rephrase to ’50 of these’Line 297: “kiringing” correct to “kriging” (I think?)Line 303 missing figure number.Line 358: suggest deleting this phrase which does not make sense to me “during their reproductive cycle end” since the paper covers anaemia among pregnant women.Line 371: this sentence is unclear please change: “Moreover , the number of cluster differences between EDHS has its effect[10, 28].”Line 394 please replace “women who were rich have less likely” with ‘women who were rich are less likely’Line 396-403 – many grammatical errors here, please check and correct.Line 419 please correct the grammar in this sentence: “However, being women having more than 1 child within five years had a positive relationship with anemia among WRA in different regions of Ethiopia.” To “However, women who had more than 1 child within five years were more likely to have anemia in multiple regions of Ethiopia.”Line 434: please change “Therefore, the Ethiopia FMoH is important to pay special” to “Therefore, it is important that the Ethiopia FMoH pay special”Line 437-438 please correct “should be a strength” to “should be strengthened”.Fig 1 define HghFigures 2, 3 correct title: trend of anaemia ‘in’ reproductive women.Figure 4 spell check title.Figure 7 – no cold spots are visible.Figure 8 and 9 seem to be duplicates?Please submit your revised manuscript by Feb 19 2022 11:59PM. If you will need more time than this to complete your revisions, please reply to this message or contact the journal office at plosone@plos.org. Please include the following items when submitting your revised manuscript:A rebuttal letter that responds to each point raised by the academic editor and reviewer(s). You should upload this letter as a separate file labeled 'Response to Reviewers'.A marked-up copy of your manuscript that highlights changes made to the original version. You should upload this as a separate file labeled 'Revised Manuscript with Track Changes'.An unmarked version of your revised paper without tracked changes. You should upload this as a separate file labeled 'Manuscript'.If applicable, we recommend that you deposit your laboratory protocols in protocols.io to enhance the reproducibility of your results. Protocols.io assigns your protocol its own identifier (DOI) so that it can be cited independently in the future. For instructions see: https://journals.plos.org/plosone/s/submission-guidelines#loc-laboratory-protocols. Additionally, PLOS ONE offers an option for publishing peer-reviewed Lab Protocol articles, which describe protocols hosted on protocols.io. Read more information on sharing protocols at https://plos.org/protocols?utm_medium=editorial-email&utm_source=authorletters&utm_campaign=protocols.

We look forward to receiving your revised manuscript.

Kind regards,

Lucy C. Okell

Academic Editor

PLOS ONE
---

## [Author Response · Author response to Decision Letter 1]

14 Jan 2022

Date: January 05/ 2022

Response to editor’s comment 

Manuscript title: Geographically weighted regression analysis of anemia and its associated factors among reproductive age women in Ethiopia using the 2016 Demographic and Health Survey.

Manuscript number: PONE-D-20-35910 

Dear editor, thank you for the most important issues you raised for the betterment of our manuscript. Below are the response for the comments and concerns you raised. In addition, we have incorporated the comments and concerns in the revised manuscript. 

 Response to editor’s comments 

Introduction section

• There is an error in the anemia units. The definition should be 120 g/L not g/dL.

Authors’ response: Dear editor thank you for your important comments. We revised it as hemoglobin level <11g/dl for lactating or pregnant mothers and hemoglobin level <12 g/dl and for none pregnant or non-lactating women which means 120 g/L.

• Please run a full spelling check on the manuscript. E.g. see typos in the abstract: “identifay” -> identify and main text “pridictors” -> predictors.

Authors’ response: Dear editor thank you for your concern. We have cheeked all the manuscript about typographic errors and we gave a serious concern and amend in all the manuscript sections.

• Line 50 suggest rewording “Reproductive age women are women who are found in the age from 15 up to 49 full-term years”to:Reproductive age is commonly defined among women as ages 15 to 49 years.

• Line 57: suggest rewording: “It is one of the most serious dangers to children's health and a factor in maternal mortality, because it increases the risk of adverse pregnancy outcomes, child mortality, impaired neurocognitive abilities, and physical development of children, and reduces work capacity despite being straightforward to prevent and treat.

Authors’ response: Dear editor thank you for your comments. We have revised and correct all the comments you raised in line 50 and 57 based on your suggestions. Line 49…and line 57.

• Line 69: “the pooled prevalence of anemia in pregnant women was 31.66% in Ethiopia” Do you have the year for this metric?

Authors’ response: Dear editor thank you for your comments. Corrected as “The pooled prevalence of anemia among pregnant women from 2003 to 2016 was 31.66% in Ethiopia”. Line 68-69.

• Line 70: replace “lactating women” with ‘current lactation’

• Line 73: small ‘a’ for “Anemia’ – please correct this throughout the text.

• Line 83: “supplying”? (not supplaying)

Authors’ response: Dear editor thank you for your suggestion. Line 70, 73 and 83 corrected accordingly in the revised manuscript.

Methods

• Line 104: “de juries” please define.

Authors’ response: Dear editor thank you for your suggestion. Women who were usually live in the surveyed households are known as de juries. defined in line 104.

• Line 118 “khat” please define. 

Authors’ response: Dear editor thank. Khat chewing is chewing of stimulant leaves, which means it speeds up the messages going between the brain and the body. Defined in revised manuscript line 118 as “stimulant plant”.

• Line 126 “we recode them based on the appropriate measure of central tendency” This is unclear – what does it mean? Centralise / normalise based on mean and SD?

Authors’ response: Dear editor thank you for your comments. We revised as “The normal distribution of aggregated community factors was assessed by histogram and Shapiro Wilks test but, they didn’t fulfill the normality assumption then we recode them based on the median value”. Line 125-128.

• Line 149: “used for parameter estimation” please change to ‘included in the model’.

• Line 150: “among reproductive age between clusters” Please change to “among reproductive age women between clusters.

Authors’ response: Dear editor thank you for your comments. Line 149-150 are revised according to your suggestions.

• Line 151. This does not quite make sense as the metrics you are mentioning are not the same as random effects. Please reword “It was estimated by…” to “We estimated the…”

Authors’ response: Dear editor thank you for your comments. Revised as “We used cluster number variable (v001) for random effect estimates. We estimated the intraclass correlation coefficient (ICC)………,.” Line 153-155.

• Line 156: please change “that would have if” to ‘associated with’.

• Line 163: I did not follow this sentence, I think it could be removed: “In this study, since the ICC was greater than 5% and the MOR have significant difference, doing the multilevel analysis was necessary” as the methods are already clear.

Authors’ response: Dear editor thank you for your comments. Revised accordingly line 156 rephrased and line 163 removed.

• Line 200. Sentence seems incomplete: “Unlike OLS that fits a single linear regression equation to all of the data in the study area, GWR 200 creates an equation for each.” …. Each?

Authors’ response: Dear editor thank you. Completed as “for each coefficient”.

Result

• Table 1: please indicate what the numbers in brackets are.

Authors’ response: Dear editor thank you. Revised as percentage (%)

• Line 233: this sentence needs rewording, grammar is incorrect and meaning unclear: “The MOR value in the null model also showed that, between higher and lower odd clusters anemic among reproductive-age women were different by 2.28 times.”

Authors’ response: Dear editor thank you. Paraphrased as “The MOR value in the null model also showed that, anemic among reproductive-age women were different by 2.28 times between higher and lower prevalence clusters”.

• Fig 3: Pia chart – do you mean Pie chart?

• Line 236 replace “expressed” with ‘explained’.

• Line 286: “from these 50 of these” please rephrase to ’50 of these’

• Line 297: “kiringing” correct to “kriging” (I think?)

• Line 303 missing figure number.

Authors’ response: Dear editor thank you for your comments. All the above comments revised and paraphrased accordingly.

Discussion

• Line 358: suggest deleting this phrase which does not make sense to me “during their reproductive cycle end” since the paper covers anaemia among pregnant women.

Authors’ response: Dear editor thank you for your comments. Deleted the phrase.

• Line 371: this sentence is unclear please change: “Moreover, the number of cluster differences between EDHS has its effect[10, 28].”

Authors’ response: Dear editor thank you for your comments. This is to explain the sample size difference might have effect on prevalence. Revised as “Moreover, the number women included in each EDHS might have its own effect”

• Line 394 please replace “women who were rich have less likely” with ‘women who were rich are less likely’- Corrected accordingly.

• Line 396-403 – many grammatical errors here, please check and correct.

Authors’ response: Dear editor thank you. Corrected as “This could be due to that, when a woman has improved wealth status, she enables to purchase healthy nutrition, and can utilize health services”

• Line 419 please correct the grammar in this sentence: “However, being women having more than 1 child within five years had a positive relationship with anemia among WRA in different regions of Ethiopia.” To “However, women who had more than 1 child within five years were more likely to have anemia in multiple regions of Ethiopia.”

• Line 434: please change “Therefore, the Ethiopia FMoH is important to pay special” to “Therefore, it is important that the Ethiopia FMoH pay special”

• Line 437-438 please correct “should be a strength” to “should be strengthened”.

Authors’ response: Dear editor thank you for your paraphrasing. Line 419,434, and 437 are corrected according to your suggestion.

• Fig 1 define Hgh

Authors’ response: Dear editor thank you for concern. We revised the figure and define all the terms. HGB- Hemoglobin, WRA- Women in Reproductive Age, and De jure - Women who usually live in the surveyed households.

• Figures 2, 3 correct title: trend of anaemia ‘in’ reproductive women.

• Figure 4 spell check title.

Authors’ response: Dear editor thank you for your comment. Spell cheeked and corrected accordingly.

• Figure 7 – no cold spots are visible.

Authors’ response: Dear editor thank you for your comment. There are significant cold spot areas of 90% confidence interval, but there were no significant cold spot area of 95% and 99% confidence interval.

• Figure 8 and 9 seem to be duplicates?

Authors’ response: Dear editor thank you for your comment. Yes you are correct, the image 8 was error. We revised the and plot the new image 8

---

## [Decision Letter · Decision Letter 2]

9 Sep 2022

Geographically weighted regression analysis of anemia and its associated factors among reproductive age women in Ethiopia using the 2016 Demographic and Health Survey

PONE-D-20-35910R2

Dear Dr. Belay,

We’re pleased to inform you that your manuscript has been judged scientifically suitable for publication and will be formally accepted for publication once it meets all outstanding technical requirements.

Kind regards,

Hubert Amu

Academic Editor

PLOS ONE

Additional Editor Comments (optional):

I recommend acceptance of this manuscript as also recommended by the reviewer of your revised manuscript.

Reviewers' comments:

Reviewer's Responses to Questions

**Comments to the Author**

1. If the authors have adequately addressed your comments raised in a previous round of review and you feel that this manuscript is now acceptable for publication, you may indicate that here to bypass the “Comments to the Author” section, enter your conflict of interest statement in the “Confidential to Editor” section, and submit your "Accept" recommendation.

Reviewer #4: All comments have been addressed

2. Is the manuscript technically sound, and do the data support the conclusions?

Reviewer #4: Yes

3. Has the statistical analysis been performed appropriately and rigorously? 

Reviewer #4: Yes

4. Have the authors made all data underlying the findings in their manuscript fully available?

Reviewer #4: Yes

5. Is the manuscript presented in an intelligible fashion and written in standard English?

Reviewer #4: Yes

6. Review Comments to the Author

Reviewer #4: All comments have been addressed. However, please consider the additional comments below.

1. kindly consider changing the word "struggling" in line 82 to a more suitable wording to make the statement less judgmental. you can consider using "making efforts".

2. the Word "Married" in table 2 has been spelt wrongly as "Merried". kindly make the necessary corrections.

7. PLOS authors have the option to publish the peer review history of their article (what does this mean?). If published, this will include your full peer review and any attached files.

Reviewer #4: **Yes: **Philip Kofie

---

## [Editor Report · Acceptance letter]

13 Sep 2022

PONE-D-20-35910R2 

Geographically weighted regression analysis of anemia and its associated factors among reproductive age women in Ethiopia using 2016 Demographic and Health Survey. 

Dear Dr. Belay:

I'm pleased to inform you that your manuscript has been deemed suitable for publication in PLOS ONE. Congratulations! Your manuscript is now with our production department. 

Kind regards, 

on behalf of

Dr. Hubert Amu 

Academic Editor

PLOS ONE